# Gain-of-function genetic screening identifies the antiviral function of TMEM120A via STING activation

Shuo Li [1,2,4], Nianchao Qian [1,2,4], Chao Jiang[1,2], Wenhong Zu[1], Anthony Liang[3], Mamie Li [3], Stephen J. Elledge [3] & Xu Tan [1,2✉]

Zika virus (ZIKV) infection can be associated with neurological pathologies, such as microcephaly in newborns and Guillain-Barre syndrome in adults. Effective therapeutics are currently not available. As such, a comprehensive understanding of virus-host interactions may guide the development of medications for ZIKV. Here we report a human genome-wide overexpression screen to identify host factors that regulate ZIKV infection and find TMEM120A as a ZIKV restriction factor. TMEM120A overexpression significantly inhibits ZIKV replication, while TMEM120A knockdown increases ZIKV infection in cell lines. Moreover, Tmem120a knockout in mice facilitates ZIKV infection in primary mouse embryonic fibroblasts (MEF) cells. Mechanistically, the antiviral activity of TMEM120A is dependent on STING, as TMEM120A interacts with STING, promotes the translocation of STING from the endoplasmic reticulum (ER) to ER-Golgi intermediate compartment (ERGIC) and enhances the phosphorylation of downstream TBK1 and IRF3, resulting in the expression of multiple antiviral cytokines and interferon-stimulated genes. In summary, our gain-of-function screening identifies TMEM120A as a key activator of the antiviral signaling of STING.

---

[1] Beijing Advanced Innovation Center for Structural Biology, Beijing Frontier Research Center for Biological Structure, MOE Key Laboratory of Bioorganic Phosphorus Chemistry & Chemical Biology, School of Pharmaceutical Sciences, Tsinghua University, Beijing 100084, China. [2] Tsinghua-Peking Center for Life Sciences, Beijing 100084, China. [3] Division of Genetics, Brigham and Women's Hospital, Howard Hughes Medical Institute, Department of Genetics, Harvard Medical School, Boston, MA 02120, USA. [4] These authors contributed equally: Shuo Li, Nianchao Qian. ✉email: xutan@tsinghua.edu.cn

Zika virus (ZIKV) is a mosquito-borne, enveloped positive-strand RNA flavivirus in the Flaviviridae family[1,2]. ZIKV was first isolated from monkeys in the Zika forest, Uganda, in 1947[3] and was known to cause limited, sporadic outbreaks in Africa and Southeast Asia over the past few decades[4]. Previously, it was believed that infected individuals showed mild symptoms or did not develop disease[5]. However, increasing evidence has shown that ZIKV can cause fetal microcephaly in newborns and Guillain-Barre syndrome in adults after the most recent outbreak in Brazil in 2015[6–8]. Worldwide attention has been drawn to the outbreak, and great progress has subsequently been made in understanding ZIKV biology, the ability to diagnosis infection, and the development of vaccines[9,10]. However, currently no approved vaccine or drug for the prevention or treatment of ZIKV infection exists[11].

During viral infection, a large number of host proteins are hijacked or disrupted to facilitate the production of new virions. Meanwhile, host proteins with antiviral functions (restriction factors) are mobilized to defend against virus infection, in part through activation of the interferon signaling pathway. Therefore, a comprehensive understanding of the virus-host interactions at the molecular level is crucial to developing novel therapeutic strategies against viruses. Using loss-of-function genome-wide screens based on CRISPR-Cas9 and RNA interference (RNAi) libraries, dependency factors including *oligosaccharyltransferase (OST) complex*[12], *endoplasmic reticulum membrane complex (EMC)*[13], *endoplasmic reticulum-associated signal peptidase complex (SPCS) proteins*[14], *integrin αvβ5*[15], and *restriction factors IFN-α-inducible protein 6 (IFI6)*[16] have been identified for ZIKV. Based on transcription profiling, *aryl hydrocarbon receptor (AHR)* was identified as a host factor for ZIKV replication[5]. Furthermore, proteomic analysis and ZIKV-host protein interactomes identified *neural cell adhesion molecule (NCAM1)* as a potential ZIKV receptor[17], interaction between ZIKV NS4A and *ANKLE2*, which is a gene linked to hereditary microcephaly[18], and *Dicer* as a ZIKV restriction factor[19]. Remarkable progress has been made using the methods above in the past few years. However, loss of function screens clearly cannot identify all relevant factors, as each screen is performed in a particular cell line that may lack expression of particular relevant factors or has redundancy of such factors. Gain-of-function screens using ORF libraries can overcome these limitations, yet few such gain-of-function screens have been reported to date.

In this study, we utilize a recently developed genome-wide lentiviral open reading frame (ORF) overexpression library to screen for host factors related to ZIKV infection. We identify a ZIKV restriction factor TMEM120A and reveal that it functions as a key activator of STING, potentiating STING-dependent innate immune responses to inhibit multiple RNA and DNA viruses.

## Results

### Identification of ZIKV restriction factors using human genome-wide lentiviral ORFs overexpression screen.
To identify host restriction factors for ZIKV infection, we performed a gain-of-function screen in Huh7 human hepatoma cells using a pooled human genome-wide lentiviral ORFs library encoding more than 16,000 human proteins[20,21]. Each ORF in the library is barcoded with 25 base-pairs random sequences, whose identity has been pre-determined with deep sequencing. Huh7 cells, which are permissive to ZIKV infection[22], were transduced with the lentiviral ORFs library (400 million cells at MOI = 0.2), selected with puromycin, and then infected with ZIKV at an MOI of 5. On day 10 postinfection, approximately 95% of cells died from ZIKV induced cytopathic effects. We performed the screen in duplicate

and extracted the genomic DNA from the surviving cells, and amplified the barcodes for next-generation sequencing (Fig. 1a). Normalized reads of ZIKV infected cells were compared to mock-infected cells, which were grown and processed in parallel. Candidate ZIKV restriction factors were identified based on an enrichment threshold of log2 reads count >10 and log2 fold of change (Log2FC) > 1 (Fig. 1b and Supplementary Data 1). Multiple restriction factors, including interleukin-27 subunit alpha[23] and well-known type I interferons IFNA2, IFNA4, IFNA6, and IFNA7 were statistically enriched after ZIKV infection. Furthermore, gene ontology analysis of the top 250 candidates revealed that most of these candidates are involved in antiviral processes such as TRAF6 and IRF7 activation, cytokine-mediated signaling pathway (Fig. 1c), supporting the reliability of our screen.

**TMEM120A inhibits ZIKV infection.** To validate the candidate restriction factors identified from the screen, we established Huh7 cell lines stably expressing each candidate ORFs from the top hits. Among these hits, the previously reported IL27 and IFNA2 proteins[24], and a protein TEPP, inhibited ZIKV infection in Huh7 cells (Fig. 1d–f). Since ZIKV preferentially infects neural stem/progenitor cells and immature neurons and causes microcephaly and other brain abnormalities in infants infected in utero[15], we also confirmed the effect of IFNA2/IL27/TEPP on ZIKV infection in a glioma cell line U87MG (Supplementary Fig. 1).

From the list of screen hits, we focused on a highly conserved candidate, TMEM120A, also known as TACAN, NET29, or TMPIT, which has been reported to be primarily expressed in the heart, kidneys, colon, and sensory neurons of the dorsal root ganglia (DRGs)[25,26]. We found that ectopic expression of TMEM120A significantly suppressed ZIKV infection in both Huh7 and U87MG cells (Fig. 2a–e). To complement these overexpression studies, we established TMEM120A knockdown cell lines using shRNAs targeting TMEM120A. TMEM120A knockdown indeed increased ZIKV infection in both Huh7 and U87MG cells (Fig. 2f, g, Supplementary Fig. 2) showing that it is rate-limiting for viral function, much like a previous factor we discovered, IFITM3[27]. Besides ZIKV, TMEM120A also inhibited infection by dengue virus (DENV), yellow fever virus (YFV), two members of the Flavivirus genus (Supplementary Fig. 3), and herpes simplex virus type 1 (HSV-1), a double-stranded DNA virus (Supplementary Fig. 6a, b) in U87MG cells.

The predicted membrane topology suggests that TMEM120A has an amino-terminal domain and six transmembrane domains with a short carboxyl-terminal tail[26]. To investigate which part of TMEM120A is essential for its antiviral activity, we constructed two truncations, named NTD (N-terminal domain, aa 1–135) and CTD (C-terminal domain, aa 136–343) (Fig. 2h) which includes the six transmembrane domains. We found that CTD inhibited ZIKV infection to a similar level as the full-length TMEM120A (Fig. 2i, j), indicating the importance of CTD for the antiviral effect.

**TMEM120A inhibits ZIKV replication independent of its proposed ion channel function.** To determine which step of the ZIKV life cycle is affected by TMEM120A, we first examined the cellular entry of ZIKV and found no significant decrease in ZIKV entry due to TMEM120A overexpression in Huh7 cells (Fig. 3a). Previous studies reported localization of TMEM120A on the nuclear membrane[28] or plasma membrane[26]. We also confirmed the localization of TMEM120A in HEK293T cells and found that TMEM120A is also localized to the endoplasmic reticulum (ER) as TMEM120A co-localizes with calnexin which is an ER marker (Fig. 3b, c). Because ZIKV initiates replication including viral

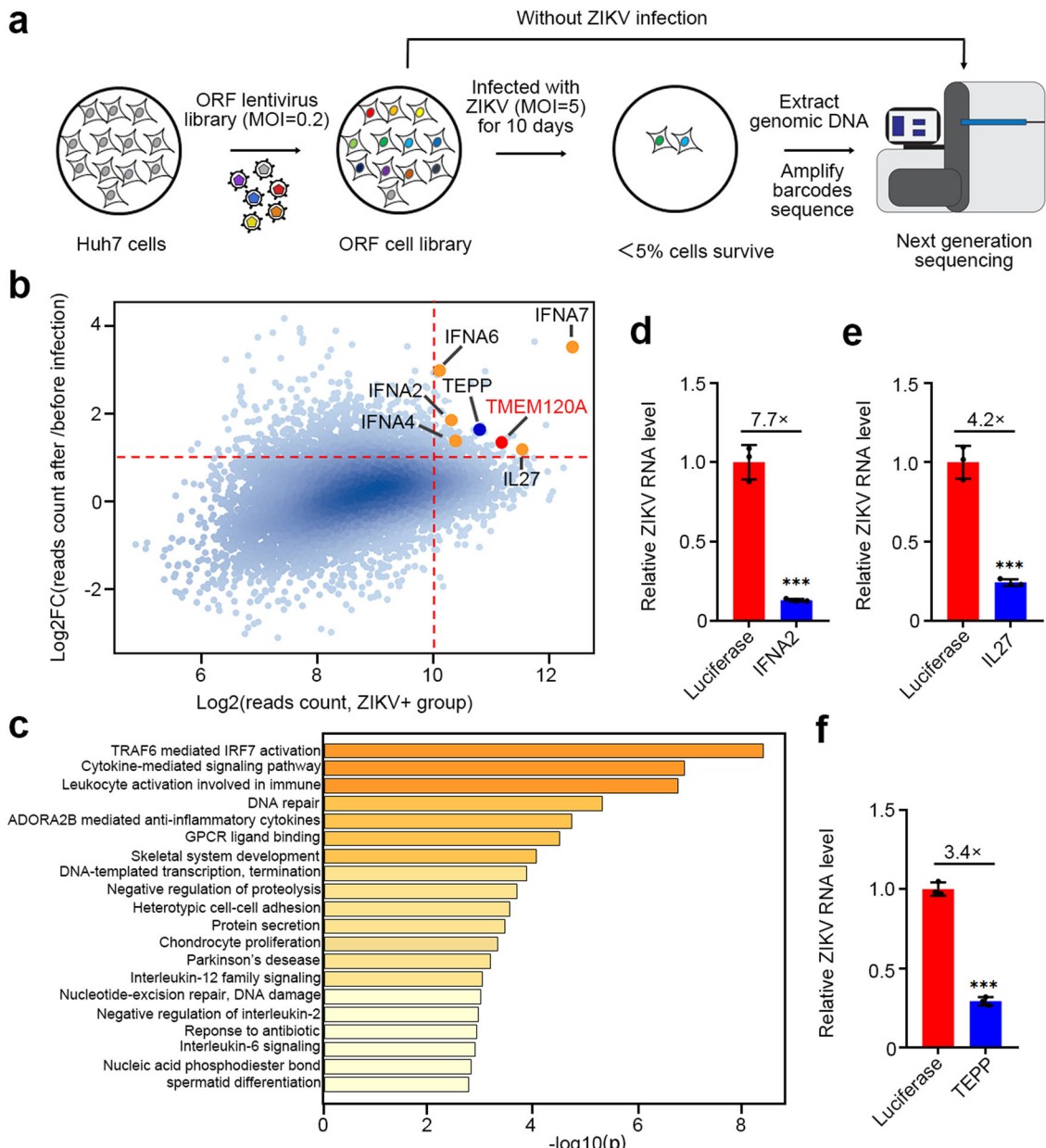

**Fig. 1 Human genome-wide gain-of-function screening identifies host factors regulating ZIKV infection in Huh7 cells. a** Schematic of human genome-wide overexpression screen. **b** Scatter plot of screen candidates. The x-axis is log2 reads count collected from ZIKV infected cells, and the y-axis is log2 fold changes (FC) in reads count collected from ZIKV infected cells relative to uninfected cells. ORFs with Log2FC >1 and log2 (reads count) >10 (Red dotted lines) were considered for validation. Reported restriction factors IL27 and well-known type I interferons IFNA2, IFNA4, IFNA6, and IFNA7 are labelled with orange dots. ZIKV restriction factors TEPP and TMEM120A are labelled with blue and red dots, respectively. **c** Gene ontology biological processes analysis of the top 250 candidates in Supplementary Data 1 using Metascape webserver. **d**–**f** Overexpression of IFNA2 (**d**), IL27 (**e**), TEPP (**f**) significantly reduced ZIKV RNA level in Huh7 cells, respectively. Huh7 cells stably expressing HA-FLAG-tagged luciferase or IFNA2, IL27, TEPP were infected with ZIKV at an MOI of 0.1 for 48 h. Cells were then harvested for RT-qPCR. Cellular ZIKV RNA level was normalized to the internal control GAPDH. RT-qPCR data represent the mean ± SEM ($n = 3$ independent experiments). ***$P < 0.001$ (unpaired, two-sided Student's t-test). Exact P values and statistical parameters are provided in Source Data File.

RNA and protein synthesis on the ER[29], we hypothesized that TMEM120A may affect ZIKV replication. To test this, we detected viral replication using a ZIKV replicon assay with a luciferase reporter[30]. At 48 h posttransfection, TMEM120A overexpression significantly inhibited luciferase activity of the replicon, suggesting TMEM120A inhibits ZIKV replication (Fig. 3d).

Recently, TMEM120A has been reported to be an ion channel involved in sensing mechanical stimuli[26]. To investigate whether

TMEM120A inhibits ZIKV replication through its channel function, we introduced mutations in TMEM120A at amino acid residues 207, 215, 228, which have been reported to be the ion-conducting part responsible for mechanically-evoked currents[26]. However, individual or combined mutations of these three residues did not affect the antiviral functions of TMEM120A (Supplementary Fig. 4a, b). Furthermore, the antiviral effects of TMEM120A in U87MG cells were not abolished in the culture medium supplemented with different ions such as $Ca^{2+}$, $Gd^{3+}$,

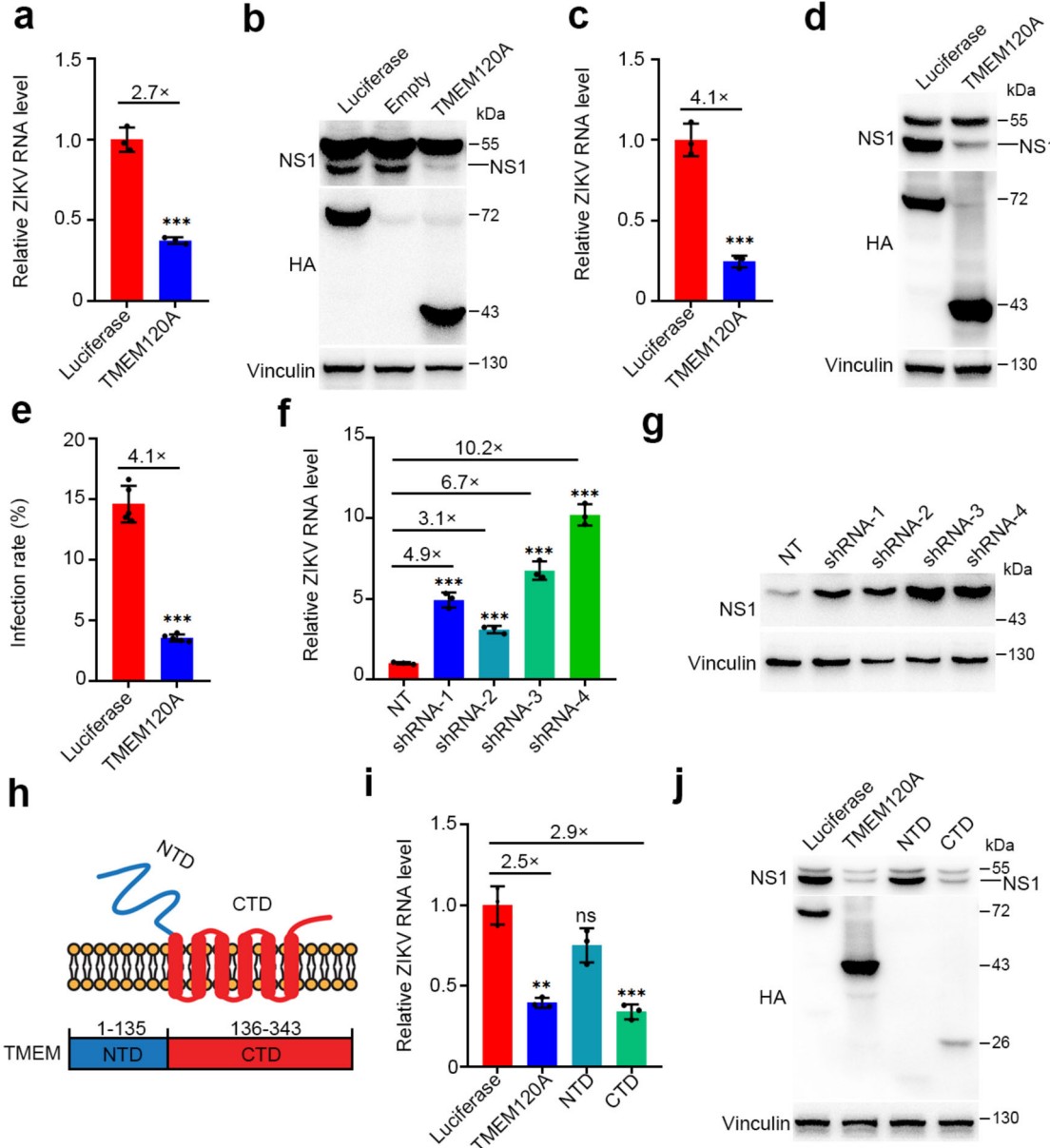

**Fig. 2 TMEM120A inhibits ZIKV infection. a**, **b** TMEM120A overexpression significantly reduced ZIKV RNA level (**a**) and NS1 protein level (**b**) in Huh7 cells. Huh7 cells stably expressing HA-FLAG-tagged luciferase or TMEM120A were infected with ZIKV at an MOI of 0.1 for 48 h. Cells were then harvested for RT-qPCR and immunoblotting. **c**, **d** TMEM120A overexpression significantly reduced ZIKV RNA level (**c**) and NS1 protein level (**d**) in U87MG cells. U87MG cells stably expressing HA-FLAG-tagged luciferase or TMEM120A were infected with ZIKV at an MOI of 0.1 for 48 h. Cells were then harvested for RT-qPCR and immunoblotting (NS1 antibody). **e** TMEM120A overexpression inhibited ZIKV infection rate in U87MG cells. U87MG cells stably expressing HA-FLAG-tagged luciferase or TMEM120A were infected with ZIKV at an MOI of 0.2 for 48 h. Cells were then fixed, permeabilized for immunostaining of ZIKV envelope protein. The infection rate was measured by high content scanning. **f** shRNA based knockdown of TMEM120A increased ZIKV RNA level in Huh7 cells. Huh7 cells stably expressing non-targeting control (NT) shRNA or TMEM120A shRNAs (four independent shRNAs: shRNA-1, shRNA-2, shRNA-3, shRNA-4) were infected with ZIKV at an MOI of 0.1 for 48 h. Cells were then harvested for RT-qPCR. **g** shRNA based knockdown of TMEM120A increased ZIKV NS1 protein level in U87MG cells. U87MG cells stably expressing NT shRNA or TMEM120A shRNAs were infected with ZIKV at an MOI of 0.1 for 48 h. Cells were then harvested for immunoblotting (NS1 antibody). **h** Domain structure of TMEM120A. NTD: N-terminal domain or cytoplasmic domain; CTD: C terminal domain or transmembrane domain. **i**, **j** TMEM120A CTD but not NTD significantly decreased ZIKV RNA level (**i**) and NS1 protein level (**j**) in U87MG cells. U87MG cells stably expressing HA-FLAG-tagged luciferase, TMEM120A, NTD, or CTD of TMEM120A were infected with ZIKV at an MOI of 0.1 for 48 h. Cells were then harvested for RT-qPCR and immunoblotting (NS1 antibody). Cellular ZIKV RNA level was normalized to the internal control GAPDH. RT-qPCR data in (**a**, **c**, **f**, **i**) represent the mean ± SEM ($n = 3$ independent experiments). ZIKV infection rate data in (**e**) represent the mean ± SEM ($n = 5$ independent experiments). **$*P < 0.01$, ***$P < 0.001$ (**a**, **c**, **e**: unpaired, two-sided Student's $t$-test; **f**, **i**: one-way ANOVA and Dunnett's test). Exact $P$ values and statistical parameters are provided in Source Data File.

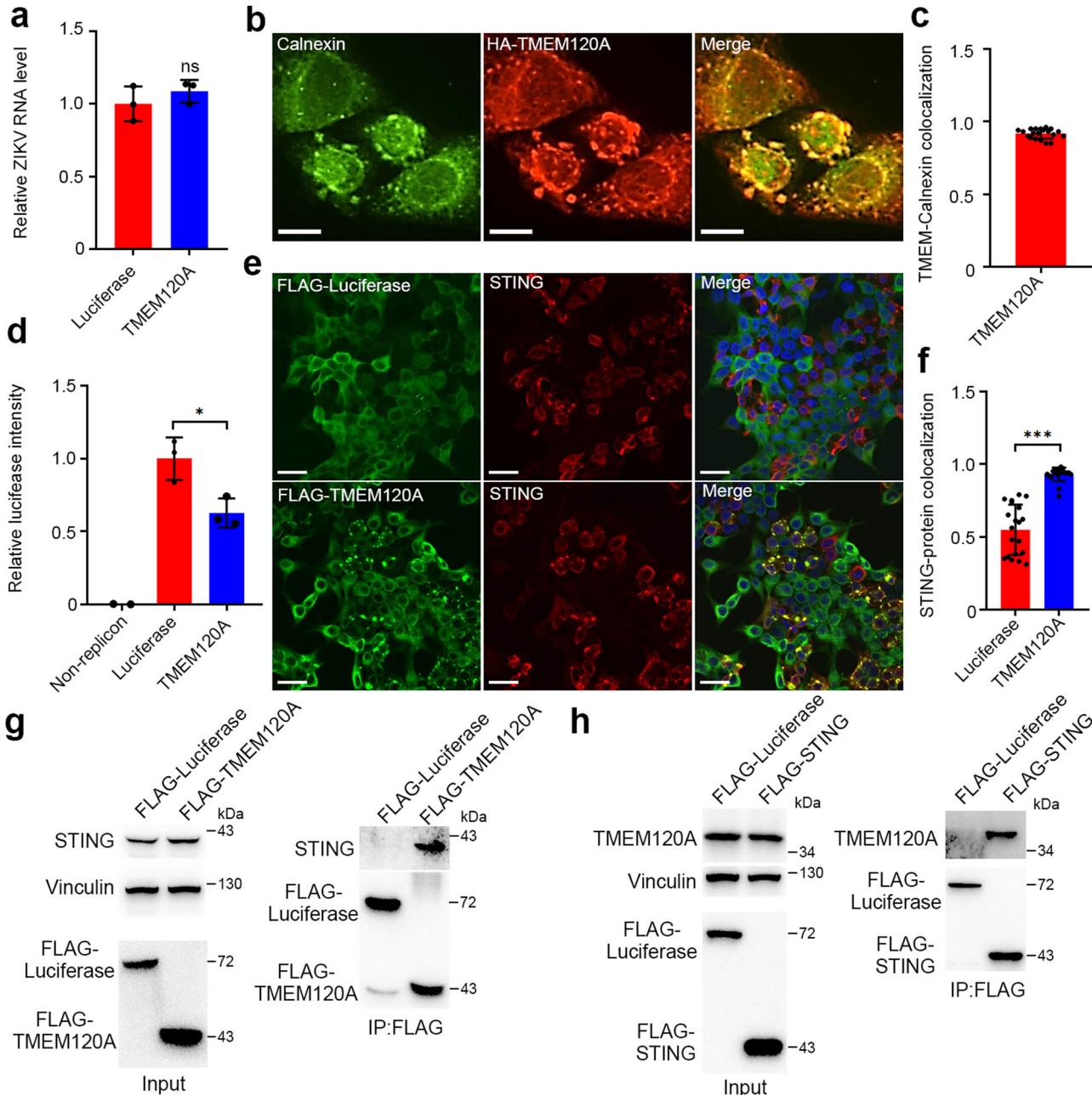

Mg$^{2+}$, Mn$^{2+}$, Cu$^{2+}$, and Zn$^{2+}$ (Supplementary Fig. 4c, d). These data suggest that the reported ion channel activity of TMEM120A does not contribute to ZIKV inhibition.

**The antiviral function of TMEM120A is dependent on STING.** STimulator of INterferon Genes (STING, also named ERIS, MITA, or TMEM173) has been identified to be essential to the innate immune response to not only DNA but also RNA viruses[31,32]. Considering that both STING and TMEM120A are localized to the ER, we wondered whether the antiviral effect of TMEM120A is associated with STING. Therefore, we first performed immunofluorescence to examine the co-localization of TMEM120A with STING in HEK293T cells. Interestingly, unlike luciferase or MAVS, TMEM120A co-localizes with a subset of STING within large foci (Fig. 3e, f and Supplementary Fig. 5). We further demonstrated that TMEM120A specifically interacts with

STING by immunoprecipitation (IP) (Fig. 3g, h), and their interaction is enhanced by ZIKV and HSV-1 infection (Supplementary Fig. 7). We also determined that TMEM120A CTD but not NTD interacts with STING by immunofluorescence and IP (Supplementary Fig. 8).

We next tested whether the interaction between TMEM120A and STING contributes to the antiviral effect of TMEM120A. First, we determined the effect of STING on ZIKV infection. We found that STING overexpression strongly suppressed ZIKV infection of U87MG glioma cells (Supplementary Fig. 9a), meanwhile STING knockdown significantly increased ZIKV infection (Supplementary Fig. 9b, c). Subsequently, we performed shRNA mediated knockdown of STING in cell lines overexpressing TMEM120A or luciferase control and infected the cells with ZIKV. We found that STING knockdown increased ZIKV infection to the same level in both luciferase and TMEM120A groups (Fig. 4a, b), indicating STING knockdown is epistatic to

**Fig. 3 TMEM120A is located at ER and associates with STING. a** TMEM120A did not inhibit ZIKV entry in Huh7 cells. Huh7 cells stably expressing luciferase or TMEM120A were incubated with ZIKV at an MOI of 1 at 37 °C for 4 h and then harvested for RT-qPCR. **b, c** TMEM120A co-localized with ER marker calnexin. HEK293T cells were transiently transfected with plasmids expressing HA-FLAG-tagged TMEM120A. At 48 h posttransfection, cells were fixed and permeabilized for immunostaining using HA (red) antibody and calnexin (green) antibody (**b**). Scale bar, 10 μm. TMEM120A-Calnexin co-localization was quantified using Pearson's correlation coefficient method. Cells expressing both Calnexin and TMEM120A were selected randomly for co-localization analysis by ImageJ software (**c**). **d** TMEM120A inhibits ZIKV replication. HEK293T cells were transfected with TMEM120A or luciferase for 24 h, then transfected with ZIKV RNA replicon. A total of 48 h post replicon transfection, cells were harvested for *Renilla* luciferase assay. **e, f** TMEM120A co-localized with STING. HEK293T cells were transiently transfected with plasmids expressing STING and HA-FLAG tagged luciferase or TMEM120A. 48 h posttransfection, cells were fixed and permeabilized for immunostaining using STING (red) antibody and FLAG (green) antibody (**e**). Scale bar, 30 μm. STING-TMEM120A or STING-luciferase co-localization was quantified using Pearson's correlation coefficient method. Cells expressing both STING and TMEM120A or luciferase were selected randomly for co-localization analysis by ImageJ software (**f**). **g** TMEM120A interacts with endogenous STING in U87MG cells. U87MG cells stably expressing HA-FLAG tagged luciferase or TMEM120A were infected with ZIKV for 48 h and then harvested for FLAG-tag based immunoprecipitation and immunoblotting to detect the interaction between TMEM120A and endogenous STING using STING antibody. 10% of the input was run and blotted. Input, STING and vinculin blots and FLAG-luciferase/TMEM120A blots are from two gels with the same samples. **h** STING interacted with TMEM120A in HEK293T cells. HEK293T cells were transiently transfected with plasmids expressing N-terminal HA-FLAG tagged luciferase or STING and C-terminal V5-TMEM120A for 48 h. Cells were then collected for FLAG-tag-based immunoprecipitation and immunoblotting to detect the interaction between TMEM120A and STING using TMEM120A antibodies. 10% of the input was run and blotted. In input, TMEM120A and vinculin binds are from one blot and FLAG-luciferase or STING band is from another blot with the same samples of the same amount of loadings. Cellular ZIKV RNA level was normalized to the internal control GAPDH. RT-qPCR in (**a**) and *Renilla* luciferase data in (**d**) represent the mean ± SEM ($n = 3$ independent experiments). Quantification of immunostaining data in (**c, f**) represent the mean ± SEM ($n = 20$ cells per group). *$P < 0.05$, ***$P < 0.001$ (unpaired, two-sided Student's $t$-test). ns: not significant. Exact $P$ values and statistical parameters are provided in Source Data File.

the antiviral phenotype of TMEM120A overproduction without affecting TMEM120A mRNA levels (Supplementary Fig. 9d, e). Since both the STING binding and antiviral function of TMEM120A depend on its CTD, these results suggest that TMEM120A may inhibit ZIKV replication in a complex with STING.

**TMEM120A activates STING signaling by promoting the ER to ERGIC translocation of STING.** Stimulated STING dissociates from the ER exit sites and traffics to the ER-Golgi intermediate compartment (ERGIC) where it recruits TBK1 to activate transcription factors IRF3 and NF-κB. These factors then activate the production of type I IFNs and inflammatory cytokines to establish an antiviral state[33]. To determine the effect of TMEM120A on the downstream signaling of STING, we first tested the trafficking of STING from the ER to ERGIC using immunofluorescence and found that more STING is co-localized with the ERGIC marker-ERGIC53 in TMEM120A overexpressing cell line compared to luciferase overexpressing cell line (Fig. 4c, d). We also found TMEM120A is co-localized with both STING and ERGIC53 (Supplementary Fig. 10). In addition, we demonstrated that TMEM120A knockdown weakened the co-localization and interaction of STING and ERGIC53 (Supplementary Fig. 11). These data suggest TMEM120A may promote STING translocation from the ER to ERGIC. As STING can be transported from ER to ERGIC via COPII-coated vesicles[34], we silenced an essential component of the COPII complex-SEC23 (encoded by two paralogous genes, SEC23A and SEC23B) using pooled siRNAs[35,36]. The result showed that knockdown of SEC23A blocked the antiviral function of TMEM120A (Fig. 4e and Supplementary Fig. 12a), presumably by decreasing the translocation of STING from the ER to ERGIC (Fig. 4f, g), suggesting TMEM120A indeed promotes STING trafficking and SEC23A plays an essential role during this process. Furthermore, we found that SEC23A knockdown did not reduce the association of TMEM120A and STING and TMEM120A did not interact with components of COPII (SEC23A, SAR1A, or SAR1B) (Supplementary Fig. 12b, c), indicating that the direct rather than COPII-mediated association of TMEM120A and STING contributes to STING trafficking.

We then measured the phosphorylation of TBK1 and IRF3, which are downstream events of STING signaling[37]. TMEM120A

overexpression increased the phosphorylation of both TBK1 and IRF3 after stimulation of STING by adding exogenous 2′, 3′-cGAMP (Fig. 4h, i) and the phosphorylation of TBK1 in response to HSV-1 infection (Supplementary Fig. 13a), supporting that TMEM120A activates STING signaling.

Next, we investigated the effect of TMEM120A on the production of type I IFNs and inflammatory cytokines by RT-qPCR in three cell lines. In HEK293T cells, we found that TMEM120A overexpression increased the mRNA levels of IFNβ, IL6, IL8, and ISGs such as ISG15, MX1 after 2′3′-cGAMP stimulation (Supplementary Fig. 14a, b). In U87MG cells, TMEM120A overexpression increased the transcription of IL6, IL8, and IFIT2 to a similar level as STING overexpression induced after ZIKV infection (Fig. 4j and Supplementary Fig. 14c). Moreover, TMEM120A knockdown significantly decreased the mRNA levels of IL6, IL8, and IFIT2 expression (Fig. 4k). In RAW 264.7 cells, which is a murine macrophage cell line, human TMEM120A overexpression also increased the transcription of cytokines and ISGs with or without HSV-1 infection (Fig. 4l and Supplementary Fig. 14d, e). These results indicated that TMEM120A indeed promotes the expression of cytokines and ISGs, which are downstream of STING signaling.

***Tmem120a* deletion in mice increased ZIKV infection in MEFs.** Given the strong antiviral activity of TMEM120A in vitro, we further tested whether TMEM120A inhibits ZIKV infection in vivo. We generated *Tmem120a* knockout (KO) mice using the CRISPR/Cas9 gene-editing system. We designed specific single-guided RNAs (sgRNAs) targeting to *Tmem120a*, and injected the sgRNAs with Cas9 mRNA into the zygotes, resulting in the deletion of a 77 base-pair (bp) fragment including the start codon ATG. Homozygous *Tmem120a* KO (*Tmem120a*$^{-/-}$) mice are lethal before embryonic day 17.5 (E17.5) (Fig. 5a and Supplementary Fig. 15a). We, therefore, used mouse embryonic fibroblasts (MEF), which are widely used for viral infection[38]. We isolated MEFs from E14.5 *Tmem120a*$^{-/-}$ mice and their wild-type (WT) littermates for further study. To validate the deletion of *Tmem120a* at the mRNA level, we designed primers targeting the deletion region of *Tmem120a* and confirmed the deletion by RT-qPCR (Supplementary Fig. 15b, c). Immunoblot experiments also confirmed the deletion in these MEFs (Fig. 5b). We infected MEFs with ZIKV or HSV-1 to determine the function of

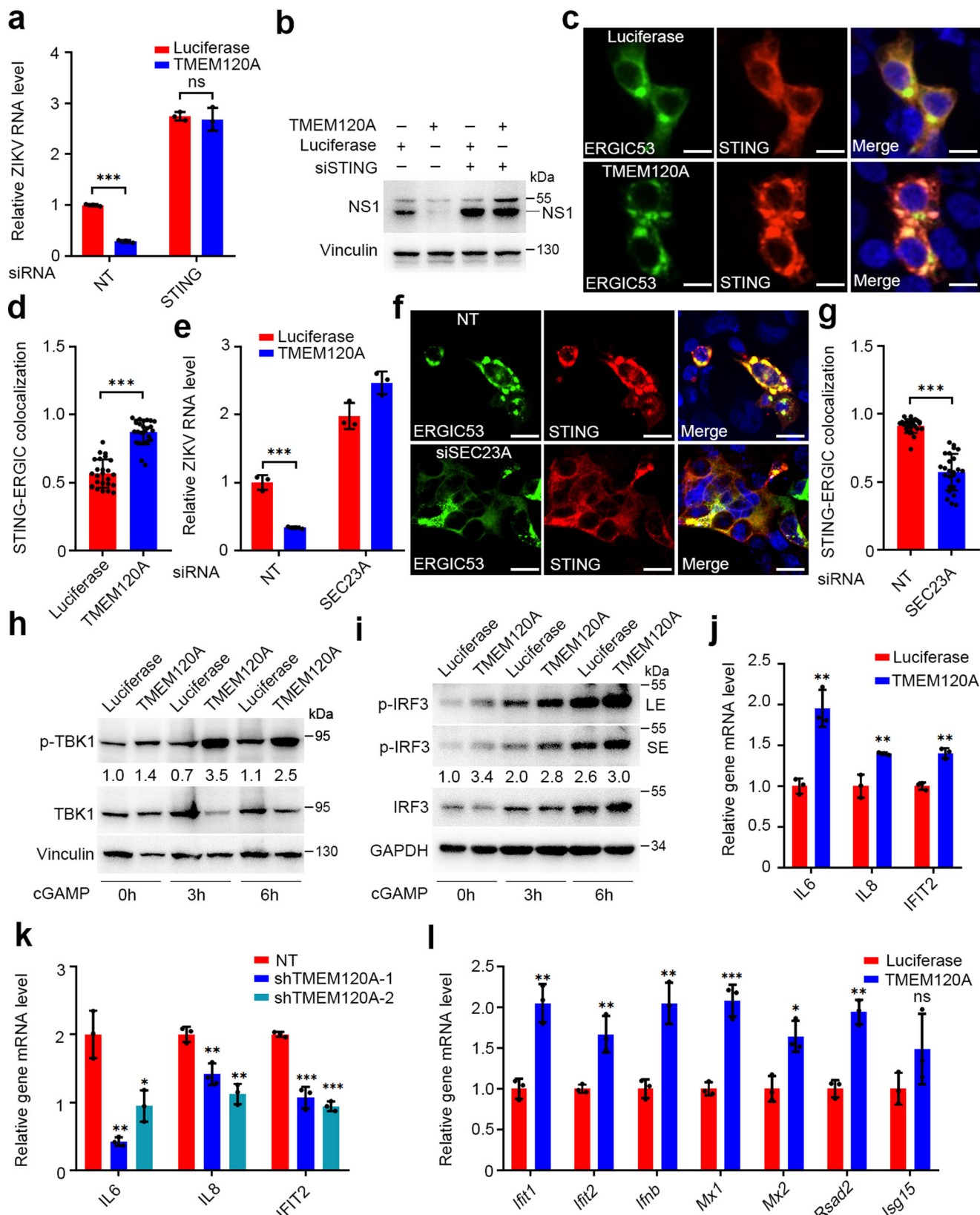

*Tmem120a* in MEFs. As expected, *Tmem120a* deletion remarkably increased ZIKV (Fig. 5c, d) or HSV-1 (Supplementary Fig. 6c, d) infection in *Tmem120a*$^{-/-}$ MEFs. We also measured the phosphorylation of Tbk1 and Irf3 in these MEFs and demonstrated that phosphorylated Tbk1 and Irf3 were reduced in *Tmem120a*$^{-/-}$ MEFs after STING activators 2′, 3′-cGAMP

(Fig. 5e, f) or DMXAA (Supplementary Fig. 15d) stimulation and the phosphorylation of Tbk1 was also impaired *Tmem120a*$^{-/-}$ MEFs in response to HSV-1 infection (Supplementary Fig. 13b). In addition, the mRNA levels of *Ifnb* and a number of *Isgs* were also decreased in *Tmem120a*$^{-/-}$ MEFs with 2′, 3′-cGAMP stimulation or ZIKV infection (Fig. 5g and Supplementary Fig. 15e).

**Fig. 4 TMEM120A promotes STING translocation and downstream signaling. a, b** siRNA-mediated STING knockdown abolished the antiviral function of TMEM120A in U87MG cells. U87MG cells stably expressing luciferase or TMEM120A were transiently transfected with non-targeting siRNA (NT) or siRNA targeting STING for 48 h and infected with ZIKV at MOI of 0.1 for another 48 h. Cells were then harvested for RT-qPCR (**a**) and immunoblotting (NS1 antibody) (**b**). **c, d** TMEM120A promoted STING trafficking from the ER to ERGIC. HEK293T cells were transiently transfected with plasmids expressing STING, GFP-ERGIC53 (green), and HA-FLAG-luciferase or TMEM120A. 48 h posttransfection, cells were fixed and permeabilized for immunostaining using STING (red) antibody and confocal analysis (**c**). Scale bar: 20 μm. STING-ERGIC53 co-localization was quantified using Pearson's correlation coefficient method. Cells expressing both STING and ERGIC53 were selected randomly for co-localization analysis by Volocity software (**d**). ERGIC53 is an ERGIC marker. **e** siRNA based SEC23A knockdown blocked the antiviral function of TMEM120A in U87MG cells. U87MG cells stably expressing Luciferase or TMEM120A were transfected with NT control siRNA or siRNA targeting SEC23A for 48 h and infected with ZIKV at an MOI of 0.1 for another 48 h. Cells were harvested for RT-qPCR. **f, g** siRNA based SEC23A knockdown decreased the translocation of STING from ER to ERGIC in HEK293T cells. HEK293T cells were transfected with NT or siRNA targeting SEC23A for 48 h, and then co-transfected with plasmids expressing STING, GFP-ERGIC53, and HA-FLAG tagged TMEM120A for another 48 h. Cells were fixed and permeabilized for immunostaining using STING (red) antibody and confocal analysis (**f**). Scale bar: 20 μm. STING-ERGIC53 co-localization was quantified using Pearson's correlation coefficient method. Cells expressing both STING and ERGIC53 were selected randomly for co-localization analysis by ImageJ software (**g**). **h, i** TMEM120A promoted TBK1 and IRF3 phosphorylation in HEK293T cells. HEK293T cells stably expressing STING were transiently transfected with HA-FLAG tagged luciferase or TMEM120A. 40 h posttransfection, these cells were stimulated with 2′, 3′-cGAMP (2 μg/mL) for the indicated time (0, 3, 6 h). Cells were collected for immunoblotting to detect the activation of TBK1 (**h**) and IRF3 (**i**) using phospho-TBK1 (p-TBK1), TBK1, p-IRF3, and IRF3 antibodies. LE, long exposure. SE, short exposure. Ratio: p-TBK1/TBK1 or p-IRF3/IRF3. **j** TMEM120A overexpression promoted type I IFN response in U87MG cells. U87MG cells stably expressing luciferase or TMEM120A were infected with ZIKV at an MOI of 0.1 for 48 h. Cells were harvested for RT-qPCR to detect the mRNA level of IL6, IL8, IFIT2. **k** shRNA based TMEM120A knockdown decreased type I IFN response in U87MG cells. U87MG cells stably expressing NT shRNA or TMEM120A shRNAs (shRNA-1, shRNA-2) were infected with ZIKV at an MOI of 0.1 for 48 h. Cells were harvested for RT-qPCR to detect the mRNA level of IL6, IL8, IFIT2. **l** TMEM120A overexpression promoted type I IFN response in RAW 264.7 cells. RAW 264.7 cells were transiently transfected with luciferase or TMEM120A and then infected with HSV-1 at an MOI of 1 for 6 h. Cells were harvested for RT-qPCR to detect the mRNA level of *Ifnb1* and other *Isgs*. Cellular ZIKV RNA level and gene mRNA level were normalized to the internal control GAPDH. RT-qPCR data in (**a, e, j, k, l**) represent the mean ± SEM (*n* = 3 independent experiments). Quantification of immunostaining data in (**d, g**) represent the mean ± SEM (*n* = 25 cells per group). \*$P < 0.05$, \*\*$P < 0.01$, \*\*\*$P < 0.001$ (unpaired, two-sided Student's *t*-test). ns: not significant. Exact *P* values and statistical parameters are provided in Source Data File.

Collectively, these data demonstrate that *Tmem120a* deletion increases ZIKV infection through attenuating innate immune responses in MEFs.

## Discussion

Following the reports of the risk of neurodevelopmental disorders in newborns caused by ZIKV infection[6–8], high-throughput screenings using CRISPR-Cas9, RNAi library, or affinity purification-mass spectrometry (AP-MS), have facilitated our understanding of host-virus interactions during ZIKV infection and provided targets for the development of innovative therapies. Given the limitation of these systems approaches, we conducted the first genome-wide gain-of-function ORF screen of ZIKV infection to explore the additional host factors in this study. The barcodes encoded *in cis* with the ORFs ensure unbiased PCR amplification and deep sequencing results, which are crucial to the accurate quantification of the abundance of the cells expressing the ORFs. We have identified several interferon-alpha genes as the top hits, validating the methodology. We further validated several genes with previously unknown anti-ZIKV functions, including IL27, TEPP, and TMEM120A. In addition, we established TMEM120A as a ZIKV restriction factor by forming a functional signaling complex with STING and promoting the trafficking of STING from the ER to ERGIC. Consequently, TBK1 is activated and phosphorylates the transcription factor IRF3, leading to activation of type I IFN expression (Fig. 6).

STING signaling protects host cells against a large number of pathogens such as DNA, RNA viruses, and also against tumor formation, while dysregulated STING also leads to autoinflammatory diseases[39]. It has previously been reported that ZIKV NS2B3 protease can cleave human STING[40]. And ZIKV infection reduced the protein level of STING in luciferase but not TMEM120A overexpressing U87MG cells in our study (Supplementary Fig. 16a). Moreover, TMEM120A overexpression protected STING protein from being cleaved by NS2B3 protease without affecting STING mRNA level in HEK293T cells overexpressing STING (Supplementary Fig. 16b, c), suggesting that

the association between TMEM120A and STING may at least partially contribute to the stabilization of STING level during ZIKV infection. However, Endogenous TMEM120A can still inhibit ZIKV infection in MEFs (Fig. 5c, d), even though NS2B3 cannot cleave mouse STING as previously reported (Supplementary Fig. 16d)[40]. Therefore, TMEM120A's stabilization of STING level is not essential for the antiviral activity.

STING signaling is strictly regulated by a variety of host factors. For example, post-translational modifications such as phosphorylation[41,42] and ubiquitination[43–48] can regulate the activation of STING signaling. Importantly, STING trafficking from ER to ERGIC/Golgi has also been reported to be regulated by several proteins including iRhom2[49], SNX8[50], TMEM203[51], and STEEP[34]. In our study, we demonstrate that TMEM120A directly interacts constitutively with STING independent of SEC23A (Supplementary Fig. 12b, c) and regulates the trafficking of STING from the ER to ERGIC to elicit antiviral effects (Fig. 4c, d, and Supplementary Figs. 10, 11). Recently, TMEM120A has also been identified to be a plasma membrane-located ion channel contributing to mechanical pain sensing in mice[26]. However, TMEM120A mutations and ions added to the culture medium have no effect on the antiviral activity of TMEM120A (Supplementary Fig. 4). These suggest that TMEM120A does not seem to regulate STING trafficking through its ion channel activity. Recent studies reported that STING exits the ER through COPII-mediated export[34]. We found that the knockdown of SEC23A, an essential component of COPII, can abolish the antiviral activity of TMEM120A (Fig. 4e). Moreover, we also demonstrated that the transmembrane domain of TMEM120A interacts with STING (Supplementary Fig. 8). Therefore, we speculate that the transmembrane domain may participate in COPII-mediated STING export through its direct interaction with STING. However, a detailed molecular mechanism remains to be fully elucidated.

Although STING signaling has been demonstrated to be important for inhibiting ZIKV infection[40], it is still unclear how STING senses the invasion of RNA viruses. Further investigation will be needed to uncover the pathway which may be also broadly

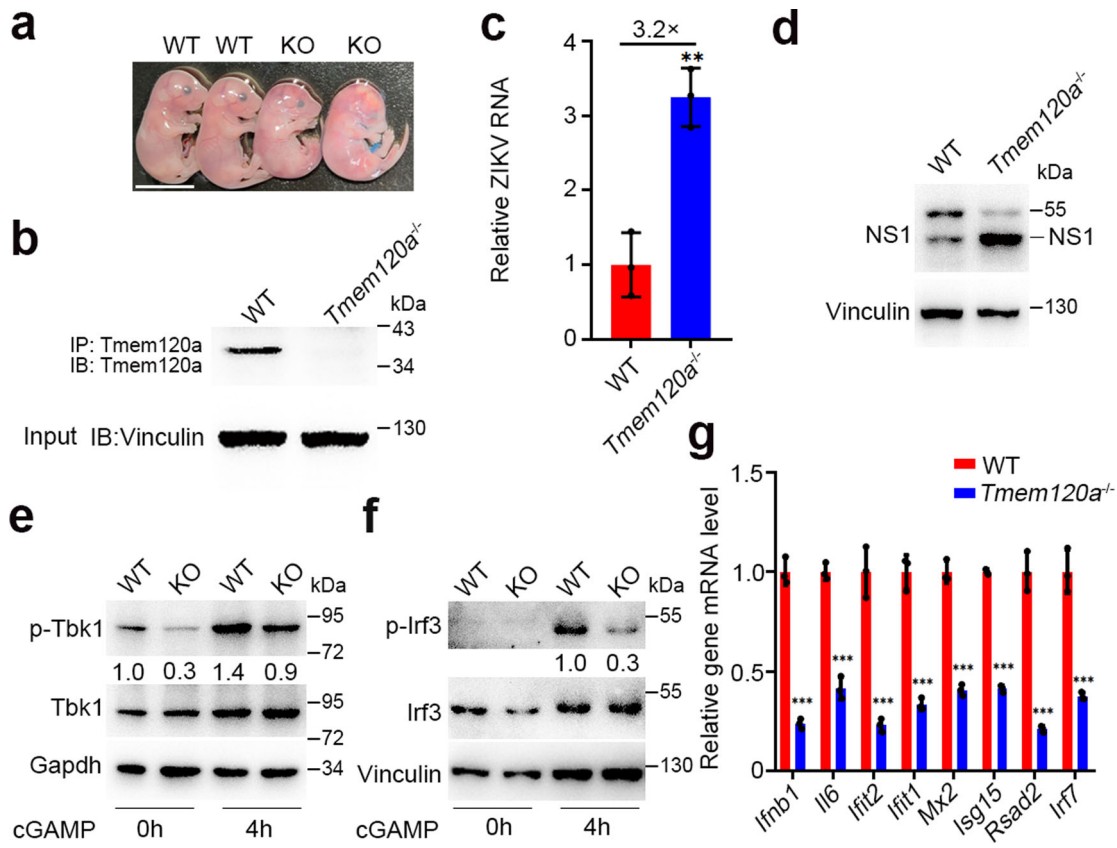

**Fig. 5 *Tmem120a*⁻/⁻ enhances ZIKV infection in MEFs. a** Photograph of WT and *Tmem120a*⁻/⁻ mouse embryos at E17.5. Scale bar: 0.5 cm. **b** Confirmation of *Tmem120a*⁻/⁻ in MEFs. WT and *Tmem120a*⁻/⁻ MEFs were harvested and *Tmem120a* was immunoprecipitated with its specific antibody followed by immunoblotting with the same antibody. The antibody we used is a polyclonal antibody targeting *Tmem120a*. 10% of the input was run and blotted. **c**, **d** *Tmem120a* deletion enhanced ZIKV infection in MEFs. WT and *Tmem120a*⁻/⁻ MEFs were infected with ZIKV at an MOI of 0.2 for 48 h. Cells were then harvested for RT-qPCR (**c**) and immunoblotting (NS1 antibody) (**d**). (**e**, **f**). *Tmem120a* deletion decreased Tbk1 and Irf3 phosphorylation in MEFs. WT and *Tmem120a*⁻/⁻ MEFs were stimulated with 2′, 3′-cGAMP (2 µg/mL) for the indicated time (0, 4 h). Cells were collected for immunoblotting to detect the activation of Tbk1 (**e**) and Irf3 (**f**) using p-Tbk1, Tbk1, pIrf3, and Irf3 antibodies. Ratio: p-Tbk1/Tbk1 or p-Irf3/Irf3. **g** *Tmem120a* KO decreases type I IFN response in MEFs. WT and *Tmem120a*⁻/⁻ MEFs were stimulated with 2′, 3′-cGAMP (2 µg/mL) for 6 h. Cells were then harvested for RT-qPCR to detect the mRNA level of *Ifnb1* and other *Isgs*. Cellular ZIKV RNA level and gene mRNA level were normalized to the internal control *Gapdh*. RT-qPCR data in (**c**, **g**) represent the mean ± SEM ($n = 3$ independent experiments). **$P < 0.01$, ***$P < 0.001$ (unpaired, two-sided Student's $t$-test). Exact $P$ values and statistical parameters are provided in Source Data File.

applicable for other RNA viruses. In summary, we demonstrate the utility of ORF-based gain-of-function screening in uncovering host factors for virus infection and establish TMEM120A as a new antiviral restriction factor and an activator of STING signaling.

## Methods

**Cell culture**. HEK293T, Huh7, U87MG, and RAW264.7 cells were cultured in Dulbecco's minimal essential medium (DMEM, Invitrogen) supplemented with 10% heat-inactivated fetal bovine serum (FBS, PAN) and 1x penicillin/streptomycin. Stable cell lines were maintained by passaging in the presence of 2 µg/mL puromycin. All cells were cultured at 37 °C, 5% $CO_2$.

**Viruses**. ZIKV (SZ01) stocks were prepared by propagation in C6 cells and titration by counting plaque-forming units (PFU). The infections of ZIKV were performed in Huh7 and U87MG cells with 0.1 MOI for 48 h. DENV was produced by passaging in Vero E6 cells. DENV infections were performed in U87MG cells with 0.1 MOI for 48 h. HSV-1(McKrae strain) was propagated in Vero E6 cells. HSV-1 infections were performed in RAW264.7 or MEFs with MOI = 1 for 48 h.

**Plasmids**. All constructs including TMEM120A, its truncations, luciferase, IL27, TEPP, STING, and IFNA2 were cloned into the pLenti-CMV, PLX304, or EF1α-FLAG lentiviral destinations via the Gateway cloning system (Invitrogen). These plasmids were then used for lentivirus production. All shRNA plasmids were purchased from Sigma.

**Antibodies**. The following antibodies were used for immunoblotting, immunoprecipitation and immunofluorescence: anti-HA (Mouse, BioLegend, *MMS-101R-200, 1:2000*), *anti*-TMEM120A (Rabbit, Proteintech, 17455-1-AP, *1:100*), anti-STING (Rabbit, Proteintech,19851-1-AP, *1:1000*), anti-STING (D2P2F, Rabbit, Cell Signaling Technologies 13647 S, *1:1000*), anti-p-IRF3 (4D4G, Rabbit, Cell Signaling Technologies, 4947, *1:500*), anti-p-TBK1 (D52C2, Rabbit, Cell SignalingTechnologies, 5483, *1:1000*), anti-IRF3 (Rabbit, Cell Signaling Technologies, 4302, *1:1000*), anti-TBK1 (Rabbit, Cell Signaling Technologies, 3504, *1:1000*), anti-GFP (Mouse, Abgent, AM1009a, *1:1000*), anti-GAPDH (Mouse, ZSGB-Bio, TA-08, *1:1000*), anti-vinculin (Mouse, Sigma, V4505, *1:5000*), anti-FLAG (Mouse, Sigma, F1084, *1:2000*), anti-Calnexin (Rabbit, Abcam, ab22595, *1:100*), Peroxidase-conjugated goat anti-rabbit antibody (ZSGB-Bio, ZB-2301, *1:20000*), Peroxidase-conjugated goat anti-mouse antibody(ZSGB-Bio, ZB-2305, *1:20000*), anti-NS1 of ZIKV (Rabbit, GeneTex, GTX133307, *1:1000*), Goat-anti-Mouse Alexa Fluor 488 (Invitrogen, A-11001, *1:1000*), Goat-anti-Rabbit Alexa Fluor 568 (Invitrogen, A-11010, *1:1000*), Goat-anti-mouse Alexa Fluor 594 (Invitrogen, A-11005, *1:1000*), Donkey-anti-Rabbit Alexa Fluor 647 (Invitrogen, A-31573, *1:1000*).

**Lentivirus packaging and transduction for establishing stable cell lines**. All lentiviruses were produced by co-transfecting 50% confluent 293 T cells with pLenti-CMV, PLX304 or EF1α-FLAG lentiviral constructs, HIV-1 gag-pol, Tat, Rev, and VSV-glycoprotein at a ratio of 10:1:1:1:2 in 6-well plates. For each transfection, 3 µL Neofect (biotech, KS2000) was combined with 3 µg total DNA in 100 µl Opti-MEM (Gibco, 31985070). A total of 6 h posttransfection, media was changed with DMEM (Gibco, 10569010) containing 10% FBS (PAN, P30-3302). Supernatants were collected at 48 h, centrifuged at 10,000 rpm, then aliquoted and stored at −80 °C until use. For lentiviral transduction, 293 T, Huh7, or U87MG cells were seeded into 6-well plates at about 50% confluence and transduced with

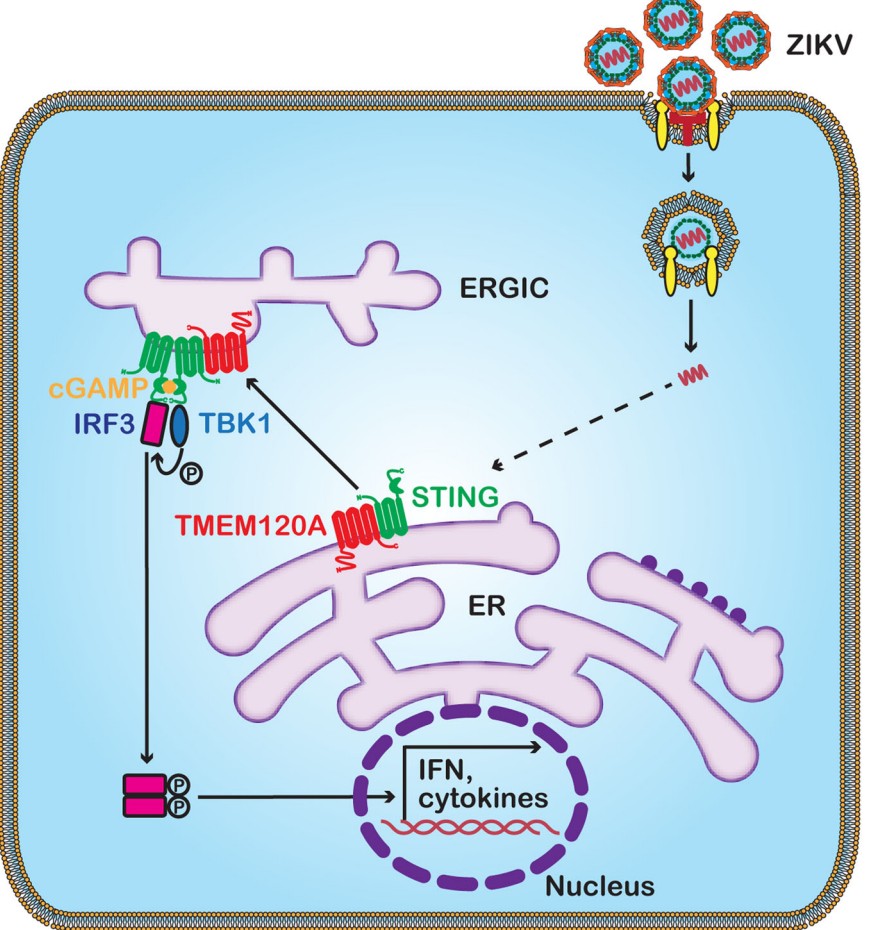

**Fig. 6 Mechanistic model of the antiviral function of TMEM120A.** TMEM120A associates with STING and promotes STING trafficking from ER to ERGIC to recruit and activate the phosphorylation of the effector kinase TBK1 and the transcription factor IRF3, thus, enhance the activation of type I IFN and cytokines expression to inhibit ZIKV replication.

1 ml packaged lentiviruses supplemented with 1 ml DMEM containing 10% FBS. 24 h posttransduction, lentiviruses were replaced with 2 ml DMEM containing 10% FBS. 48 h posttransduction, cells were selected with 2 μg/ml puromycin (Solarbio, P8230).

**Genome-wide ORF overexpression screen.** The pooled human genome-wide ORFs library contains 16,172 clonal ORFs, mapping 13,833 human genes. The genome-wide ORF overexpression screen was conducted as previously described[52]. Briefly, HEK293T cells were transfected with the ORFs library and packaging plasmids to produce the lentiviral library. Cell number Huh7 cells were transduced with the lentiviral library at 0.2 MOI to avoid a cumulative effect of multiple ORFs. Cells were then selected with 2 μg/mL puromycin for 3 days. Then the transduced cells were infected with ZIKV at MOI = 5 or mock-infected. At 10 dpi until >95% of the cells died, total genomic DNA was extracted from the survival cells using TIANGEN kits according to the manufacturer's protocol (TIANGEN BIOTECH). Barcode sequences that accompany each ORF were amplified and sequenced on a HiSeq2000 (Illumina). The resulting reads were trimmed by Cutadapt (version 1.18). Trimmed reads were mapped to the index of the ORF library using Bowtie2. Samtools (version 1.9) was used to process the alignment files to retain perfectly mapped reads only, and a custom-made Python script was used to count the reads mapped to each ORF. The reads count of ORF was then normalized using DEseq2. Log2FC of each ORF was determined relative to before infection samples. The hits with Log2FC > 1 and Log2 read count > 10 were selected as candidates. We calculated scores according to read count and fold change and sorted by scores as shown in Supplementary Data 1.

**Viral entry assay.** U87MG cells were seeded in 24-well-plate at a density of 0.05−0.1 million per well and incubated with ZIKV at MOI = 1 for 4 h at 37 °C. The supernatant was then removed and the cells were washed three times with phosphate-buffered saline (PBS, Solarbio, P1022). Total RNA was extracted followed by qRT-qPCR to measure viral entry.

**Viral replicon assay.** The plasmid containing ZIKV *Renilla* luciferase replicon was linearized by *Xho* I digestion and purified by phenol-chloroform extraction. The purified DNA was transcribed using RiboMAX T7 large-scale RNA production system to generate viral RNA (Promega, P1300). The viral RNA was purified by a HiPure Total RNA Mini Kit (Magen, R4111). Subsequently, HEK293T cells were seeded in a 24-well plate at a density of 0.05–0.1 million and transfected with plasmids expressing TMEM120A or luciferase. 24 h posttransfection, the cells were then transfected with viral RNA using *Trans*IT-mRNA transfection kit (Mirus, MIR 2225). 48 h post-viral RNA transfection, luciferase activity was measured using the *Renilla* luciferase assay system (Promega, E2810) in a Multimode plate reader (PerkinElmer).

**Quantitative real time-PCR.** Total RNA was extracted using RaPure Total RNA Micro Kit (Magen, R4111-03) according to the manufacturer's instructions. Total RNA within 2 μg was reverse transcribed using cDNA Synthesis Kit (abm, G490). All interesting transcripts were quantified using SYBR qPCR enzyme (Vazyme, Q321−02) on CFX96 real-time system (Bio-Rad). Gene expression was normalized to GAPDH or vinculin. Primer sequences are listed in Supplementary Table 1. siRNAs targeting STING and SEC23A were purchased from GenePharma and JTSBIO. Sub-confluent U87MG cells in 24-well plates were transfected with mixtures containing 20 nM siRNAs and 3 μl Lipofectamine RNAiMAX (Life Technologies, 13778150) or 1 μl lipid nanoparticles (LNP). 48 h posttransfection, the cells were used for downstream applications (viral infection). The siRNA sequences are listed in Supplementary Table 2.

**Immunoblotting.** Cells were lysed in lysis buffer containing 20 mM Tris-HCl (pH 7.5), 150 mM NaCl, 2 mM EDTA, 1% Nonidet P-40 (Sigma, FlukaN74385) and 1% complete protease inhibitor cocktail (Selleck) on ice for 30 min. Cell lysates were mixed with 5x SDS loading buffer (CWBIO, CW0027S) at a ratio of 1:4 and boiled for 10 min at 95 °C. The boiled cell lysates were electrophoresed through 10% SDS-polyacrylamide gels and the separated proteins were transferred to PVDF membranes (Merck Millipore, Germany). After being blocked in 5% dry milk/TBST for an hour at RT, the membranes were incubated with diluted primary antibodies for

2 h at RT or overnight at 4 °C. Immunoreactive bands were detected using HRP-conjugated secondary antibody and ECL substrate (DiNing, DE2001) and visualized using FluorChem imager (ProteinSimple).

**Immunoprecipitation**. Cells were harvested with trypsin/EDTA solution (Gibco, 25200072) and lysed on ice for 30mine in a lysis buffer containing 20 mM Tris-HCl (pH 7.5), 150 mM NaCl, 2 mM EDTA, 1% Nonidet P-40 (Sigma), and 1% complete protease inhibitor cocktail (Selleck). After centrifugation at 15,000 g at 4 °C for 10 min, supernatants were collected. 10% of whole-cell lysates were stored as input and others were incubated with equilibrated mouse anti-FLAG M2 magnetic beads (Sigma, M8823) overnight at 4 °C on a rotator. Beads were then washed at least three times with lysis buffer. A FLAG peptide (Selleck, B23111) was used to elute the binding proteins at RT with a constant vortex for 1 h. The eluted proteins were then detected using SDS-PAGE and immunoblotting after boiling in 1x SDS buffer.

**Immunofluorescence**. Cells were fixed in 4% paraformaldehyde (PFA) for 15 min at RT and permeabilized with 0.2% Triton X-100 for another 10 min at RT. After washing three times with PBS, cells were incubated with diluted primary antibodies for 2 h at RT or at 4 °C overnight. Cells were then washed three times with PBS and incubated with Alexa Fluor conjugated secondary antibodies (ThermoFisher) for an hour at RT. After three washes in PBS, cells were then stained with DAPI (Solarbio). At last, cells were mounted with Fluoromount-G (SouthernBiotech) and images captured using Olympus FV1000 confocal microscope (Olympus) or spinning disc confocal microscope (PerkinElmer or Andor). We used Volocity or ImageJ software for co-localization analysis.

**Generation of *Tmem120a* knockout (KO) mice using CRISPR/Cas9**. CRISPR/Cas9 mediated gene editing in mice was performed as previously described[53]. Briefly, three candidate guide RNAs (gRNAs) were designed to target exon 1 of *Tmem120a* (Gene ID: 215210) with a gRNA design tool (https://sg.idtdna.com/site/order/designtool/index/CRISPR_SEQUENCE). The gRNAs were amplified using lentiCRISPR v2 as a PCR template and in vitro transcribed using T7 RNA polymerase kit (Promega). The transcripts of the 3 gRNAs were purified using RNA kit (Magen) and mixed with Cas9 mRNA (TriLink Biotechnologies) in RNase-free water at a concentration of 50, 50, 50, 100 ng/µl. Zygote injection was performed by the Laboratory Animal Research Center at Tsinghua University according to standard protocols. C57BL/6 female mice were used as embryo donors and pseudo-pregnant foster mothers. About one hundred fertilized embryos were injected with the sgRNAs-Cas9 RNA mixture and transferred into oviducts of pseudo-pregnant female mice. For genotyping of *Tmem120a*$^{-/-}$ mice, tail or toe tissues were used to extract genomic DNA using the Puregene extraction kit (Tiangen, Venlo, Netherlands). The genotype of mice was detected by PCR amplification using specific primers targeting the exon 1 of *Tmem120a*:

Forward, 5′–GCTCTGATTGGCTGCTCCTA-3′
Reverse, 5′–CGCTCACCTGGATACCTTGG-3′

All animal experiments in this study were approved by the Institutional Animal Care and Use Committee (IACUC) at Tsinghua University (approval number: 17TX-1). All mice were housed on a 12 h light/dark cycle, at temperatures of ~18-23 °C with 40–60% humidity under specific-pathogen-free conditions.

**Statistical analysis**. Data were analyzed using GraphPad Prism 8.0 and presented as mean ± SEM unless stated otherwise. For data with two groups, student's *t*-test (paired, two-sided) was used. For data with more than two groups, a one-way analysis of variance (ANOVA) test was used and appropriate adjustments were made for multiple hypothesis testing. *P* values are denoted as follow (ns: not significant, $*P < 0.05$, $**P < 0.01$, $***P < 0.001$). Exact *P* values and statistical parameters are provided in Source Data File. All experiments were repeated at least three times unless otherwise stated.

**Reporting Summary**. Further information on research design is available in the Nature Research Reporting Summary linked to this article.

## Data availability

All data that support the findings of this study are provided in the main text, Supplementary Information, and Source Data File. Any other relevant information can be obtained from the corresponding authors upon request. Source data are provided with this paper.

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

## Acknowledgements

This work was supported by grants from the State Key Research Development Program of China to X.T. (no. 2021YFC2300203) and National Natural Science Foundation of China to S.L. (32100112), and grants from Beijing Advanced Innovation Center for Structural Biology, Beijing Frontier Research Center for Biological Structure, and Tsinghua-Peking Center for Life Sciences to X.T. S.J.E. is an investigator with the Howard Hughes Medical Institute.

## Author contributions

X.T. conceived and supervised the project; S.L. and N.Q. conducted the experiments; J.C. contributed to the mouse experiments; W.Z. contributed to data analysis; A.L. and M.L. contributed reagents; S.J.E. supervised the project and analyzed the data; N.Q., S.L., S.J.E., and X.T. wrote the paper.

## Competing interests

All authors declare no competing interests.
