## [Peer Review File · Nature Communications]

Gain-of-function genetic screening identifies the antiviral function of TMEM120A via STING activationREVIEWER COMMENTS

Reviewer #1 (Remarks to the Author):

This study employed a genome-wide overexpression screen to identify host restriction factors against Zika virus (ZIKV), which may have previously been missed in previous loss-of-function screens in specific cell types. Using this platform, the authors have identified TMEM120A as a novel restriction factor against ZIKV. Overexpression of TMEM120A reduced ZIKV replication in human cells, whereas knocking down TMEM120A led to increased infection levels. Through subsequent analysis, the authors suggest that the activity of TMEM120A depends on the STING pathway. Their results indicate that TMEM120A interacts directly with STING and colocalizes with STING at the ER-Golgi intermediate compartment (an effect that can be blocked by knockdown of vesicle protein SEC23A). Overexpression of TMEM120A corresponds with increased phosphorylation of downstream signaling proteins TBK1 and IRF3, as well as increased expression of interferon stimulated genes. Finally, the authors show that CRISPR knockout of TMEM120A in mice leads to enhanced ZIKV infection in mouse embryonic fibroblasts.

Comments

1. In addition to TMEM120A, the authors identify a novel protein TEPP that significantly inhibits ZIKV infection in Huh7 (perhaps even more so than TMEM120A); however, this protein is not investigated beyond Figure 1. The authors should clarify why this protein was excluded from subsequent analysis in favor of TMEM120A.
2. In figure 4, the authors show that overexpression of TMEM120A enhances translocation of STING to the ERGIC, and that this translocation can be blocked by knockdown of SEC23A (a component of a COPII vesicles involved in STING transport). It is conceivable that TMEM120A does not interact directly with STING, but rather interacts with SEC23A or another component of the COPII complex. Immunoprecipitation experiments should be performed (a) between STING and TMEM120A in the absence of SEC23A, and (b) between TMEM120A and SEC23A in the absence of STING (e.g. in HEK293T cells) to determine whether TMEM120A and SEC23A interact. These experiments would help elucidate whether TMEM120A is involved only in STING translocation or may be involved in trafficking of other proteins from the ER.
3. It has previously been shown that the NS2B3 protease region of ZIKV can cleave STING; however, this study includes no western blot images of STING to show how this cleavage product may vary in the presence or absence of TMEM120A. This analysis should be performed to confirm the normal cleavage function of ZIKV in these cells and determine whether TMEM120A expression promotes expression of full-length STING protein in addition to elevated RNA levels.
4. It is unclear whether TMEM120A is important for STING activity beyond the context of ZIKV infection, as the title suggests. To support this claim, the authors should perform further experiments with other activators of STING (e.g. cGAMP) in cells where TMEM120A is knocked-down or overexpressed and evaluate how STING translocation and downstream phosphorylation events are impacted.
5. How was the infection dose (MOI = 5) of ZIKV – which seems very high – chosen for the screen?
6. Fig 3d: luciferase is not a good control for the localization studies. It would be better to included proteins whose subcellular location has been well established as being ER-associate (+ ctrl) or restricted to some other organelle (e.g. MAVS with mitochondria/peroxisomes).
7. Throughout the manuscript, the authors should make quantitative statements, e.g. "...significantly suppressed ZIKV infection [X fold]"(page 6, line 5)
8. Figures 3, 4, S5, S7: Please provide the Pearson's correlation coefficient for all the colocalization

analysis included in this study.

9. Does TMEM120 expression/knock-down have any impact on other flaviviruses, in particular those that are sensitive to cGAS/STING signaling (e.g. DENV)

Reviewer #2 (Remarks to the Author):

Li et al performed an overexpression screen in Huh7 cells to identify genes that inhibit Zika virus-induced cell death. Among the genes they identified was TMEM120A, a 6-transmembrane protein with poorly known function. RNAi of TMEM120A enhanced ZIKV replication. Knockdown of STING in U87MG cells abrogated the inhibitory effect of TMEM120A overexpression on ZIKV replication. TMEM120A was found to associate with STING on the ER and promote the translocation of STING from ER to ERGIC, a step previously shown to be important for STING signaling. Tmem120A-deficient mice had elevated ZIKV RNA and slightly weaker phosphorylation of TBK1 and IRF3 in response to cGAMP treatment.

Although the identification of TMEM120A as a potential regulator of STING trafficking may be interesting, there are several major concerns about the conclusions of the paper.

- 1) The entire paper relies on infection with ZIKV, which is an RNA virus, but the major function of STING is in immune defense against DNA viruses. If TMEM120A is important for STING signaling, a DNA virus such as HSV should be tested throughout.
- 2) The effect of TMEM120A deletion on the phosphorylation of TBK1 and IRF3 in response to cGAMP was quite modest (Fig 5e and 5f), especially considering that TMEM120A deletion also impaired cytokine induction by poly[I:C] (Fig 5g), which does not activate STING. Thus, the effect of TMEM120A deletion might be non-specific (to the STING pathway).
- 3) Many cancer cell lines do not express STING. Do Huh7 cells express STING? If not, the authors' model is called into question.
- 4) Fig 3e: Is the association of STING and TMEM120A dependent on virus infection? A DNA virus should be tested in this case.
- 5) Fig 4h: the effect of TMEM120A overexpression on cGAMP-induced IRF3 phosphorylation was also very weak.

Title: Gain-of-function genetic screening identifies antiviral function of TMEM120A by activating STING

Point by point responses to reviewers' comments:

We would like to thank all the reviewers for their constructive comments on our manuscript. We have revised the manuscript to hopefully address all the concerns about our work. We believe the revised manuscript has been much improved with the added data.

Reviewer #1 (Remarks to the Author):

This study employed a genome-wide overexpression screen to identify host restriction factors against Zika virus (ZIKV), which may have previously been missed in previous loss-of-function screens in specific cell types. Using this platform, the authors have identified TMEM120A as a novel restriction factor against ZIKV. Overexpression of TMEM120A reduced ZIKV replication in human cells, whereas knocking down TMEM120A led to increased infection levels. Through subsequent analysis, the authors suggest that the activity of TMEM120A depends on the STING pathway. Their results indicate that TMEM120A interacts directly with STING and colocalizes with STING at the ER-Golgi intermediate compartment (an effect that can be blocked by knockdown of vesicle protein SEC23A). Overexpression of TMEM120A corresponds with increased phosphorylation of downstream signaling proteins TBK1 and IRF3, as well as increased expression of interferon stimulated genes. Finally, the authors show that CRISPR knockout of TMEM120A in mice leads to enhanced ZIKV infection in mouse embryonic fibroblasts.

1. In addition to TMEM120A, the authors identify a novel protein TEPP that significantly inhibits ZIKV infection in Huh7 (perhaps even more so than TMEM120A); however, this protein is not investigated beyond Figure 1. The authors should clarify why this protein was excluded from subsequent analysis in favor of TMEM120A.

Response: We thank the reviewer for summarizing the key results of our work. Regarding TEPP, it has been reported to mainly express in testis, prostate, and placenta (**Ref. S1**). The expression level of TEPP is very low in Huh7 and U87MG cell lines. We have not been able to find cell lines with reasonable TEPP expression to perform knockdown experiments to confirm the effect of endogenous TEPP on ZIKV infection. Therefore, we focused on TMEM120A instead. But we are still interested in the novel protein TEPP and will explore the molecular mechanism of its inhibitory effect on ZIKV infection in the future.

References:

Ref. S1. Bera, T. K., Hahn, Y., Lee, B. & Pastan, I. H. TEPP, a new gene specifically expressed in testis, prostate, and placenta and well conserved in chordates.

2. In figure 4, the authors show that overexpression of TMEM120A enhances translocation of STING to the ERGIC, and that this translocation can be blocked by knockdown of SEC23A (a component of a COPII vesicles involved in STING transport). It is conceivable that TMEM120A does not interact directly with STING, but rather interacts with SEC23A or another component of the COPII complex. Immunoprecipitation experiments should be performed (a) between STING and TMEM120A in the absence of SEC23A, and (b) between TMEM120A and SEC23A in the absence of STING (e.g. in HEK293T cells) to determine whether TMEM120A and SEC23A interact. These experiments would help elucidate whether TMEM120A is involved only in STING translocation or may be involved in trafficking of other proteins from the ER.

Response: We thank the reviewer for the suggestion. Following the suggestion, we have silenced SEC23A using pooled two siRNAs and investigated the interaction between TMEM120A with STING. We observed that SEC23A knockdown did not affect the interaction between TMEM120A with STING (Fig. S12b and also shown below). The results suggest a direct interaction between TMEM120A and STING. Also as recommended by the reviewer, we have now determined whether TMEM120A interacts with components of the COPII complex in HEK293T cells using co-immunoprecipitation and found that TMEM120A did not interact with COPII components: SEC23A, SAR1A or SAR1B (Fig. S12c and also shown below).

Figure S12. siRNA based SEC23A knockdown do not affect the interaction

between TMEM120A with STING.

(a) siRNA based SEC23A knockdown did not affect the interaction between TMEM120A with STING in HEK293T cells. HEK293T cells were transfected with NT or siRNA targeting SEC23A for 48h, and then transiently transfected with plasmids expressing TMEM120A and N-terminal Flag tagged STING for 48h. Cells were then collected for Flag-tag based immunoprecipitation and immunoblotting to detect the interaction between TMEM120A and STING using TMEM120A antibody. 10% of input was run and blotted.

(b) siRNA based SEC23A knockdown did not affect the interaction between TMEM120A with STING in U87MG cells. U87MG cells expressing TMEM120A and Flag-STING were transiently transfected with non-targeting control (NT) siRNA or SEC23A siRNAs. 48h post transfection, cells were then collected for Flag-tag based immunoprecipitation and immunoblotting to detect the interaction between TMEM120A and STING using TMEM120A antibody. 10% of input was run and blotted.

(c) HEK293T cells were transiently transfected with plasmids expressing Flag-GFP, STING, SEC23A, SAR1A, SAR1B or luciferase. 24h post transfection, cells were transiently transfected with plasmid expressing TMEM120A for another 48h. Cells were then collected for Flag-tag based immunoprecipitation and immunoblotting to detect the interaction between TMEM120A and STING using TMEM120A antibody. 10% of input was run and blotted.

3. It has previously been shown that the NS2B3 protease region of ZIKV can cleave STING; however, this study includes no western blot images of STING to show how this cleavage product may vary in the presence or absence of TMEM120A. This analysis should be performed to confirm the normal cleavage function of ZIKV in these cells and determine whether TMEM120A expression promotes expression of full-length STING protein in addition to elevated RNA levels.

Response: We agree with this comment. The effect of TMEM120A on ZIKV NS2B3 mediated STING cleavage should be addressed to elucidate the molecular mechanism by which TMEM120A inhibits ZIKV infection.

First, we detected STING protein level in U87MG cell lines overexpressing luciferase or TMEM120A with or without ZIKV infection. The result showed that TMEM120A expression did not promote expression of STING protein, and ZIKV infection significantly reduced STING protein level in luciferase but not TMEM120A overexpressing cell line (Fig. S16a and also shown below). There are two possibilities to explain this: 1. TMEM120A directly blocks the NS2B3 protease; 2. TMEM120A inhibits ZIKV replication and therefore reduces NS2B3 level. To determine which one contributes to the anti-ZIKV effect, we next co-transfected NS2B3 with luciferase and TMEM120A in HEK293T cells overexpressing STING. Western blot of STING showed that NS2B3 could slightly reduce STING protein level and TMEM120A overexpression blocked the reduction without affecting STING mRNA level (Fig. S16b and also shown below). This suggests that the association between TMEM120A and STING may at least partially contribute to the stabilization of STING level during ZIKV infection.

However, we further showed that TMEM120A's stabilization of STING level is not essential for the antiviral activity. This can be proved by our results of ZIKV infection in MEFs. Endogenous TMEM120A can still inhibit ZIKV infection in MEFs (Fig. 4c, d), even though NS2B3 cannot cleave mouse STING as previously reported (Fig. S16d and also shown below) (Ref. S2).

We have now expanded on this point on page 13, lines 4-15 in the revised discussion.

Figure S16. TMEM120A overexpression protects STING from cleavage by NS2B3 of ZIKV.

(a) TMEM120A overexpression did not increase STING protein level, and ZIKV infection significantly reduced STING protein level in luciferase but not TMEM120A overexpressing U87MG cells. U87MG cells stably expressing HA-Flag-tagged luciferase or TMEM120A were infected with ZIKV at an MOI of 1 for 48h. Cells were then harvested for immunoblotting (STING antibody).

(b, c) TMEM120A overexpression blocked the reduction of STING protein level induced by NS2B3 expression without affecting STING mRNA level in HEK293T cells. HEK293T cells were transiently transfected with plasmids expressing N-terminal Flag tagged luciferase or TMEM120A, HA-STING and NS2B3 for 48h. Cells were then collected for immunoblotting (HA antibody) (b) and RT-qPCR (c).

(d) ZIKV infection did not reduce Sting protein level in MEFs. MEFs were infected with ZIKV at an MOI of 1 for 48h. Cells were then harvested for immunoblotting (Sting antibody).

References:

Ref. S2. Ding, Q. *et al.* Species-specific disruption of STING-dependent antiviral cellular defenses by the Zika virus NS2B3 protease. *Proc Natl Acad Sci U S A* **115**, E6310-E6318, doi:10.1073/pnas.1803406115 (2018).

4. It is unclear whether TMEM120A is important for STING activity beyond the context of ZIKV infection, as the title suggests. To support this claim, the authors should perform further experiments with other activators of STING (e.g. cGAMP) in cells where TMEM120A is knocked-down or overexpressed and evaluate how STING translocation and downstream phosphorylation events are impacted.

Response: We appreciate the reviewer's comment. Following the comment, we stimulated WT and *Tmem120a*^{-/-} MEFs with STING activators 2', 3'-cGAMP and DMXAA and measured the phosphorylation of TBK1 and IRF3. The results showed that the phosphorylation of TBK1 and IRF3 were reduced in *Tmem120a*^{-/-} MEFs after 2', 3'-cGAMP stimulation and DMXAA stimulation (Fig. 5e, 5f and Fig. S15d). The results of DMXAA stimulation were also shown below.

Fig. S15d. *Tmem120a* deletion decreases DMXAA induced TBK1 and IRF3 phosphorylation in MEFs. WT and *Tmem120a*^{-/-} MEFs were stimulated with DMXAA (10 µg/mL) for the indicated time (0, 4h). Cells were collected for immunoblotting to detect the activation of TBK1 and IRF3 using p-TBK1, TBK1, pIRF3, and IRF3 antibodies. Ratio: p-TBK1/TBK1 or p-IRF3/IRF3.

We have also performed immunofluorescence microscopy and immunoprecipitation analysis to detect the co-localization and interaction of STING with ERGIC53 after 2', 3'-cGAMP stimulation. We demonstrated that TMEM120A knockdown significantly weakened the co-localization and interaction of STING and ERGIC53 after 2', 3'-

cGAMP stimulation (Fig. S11 and also showed below), suggesting TMEM120A knockdown reduced STING translocation.

Figure S11. TMEM120A knockdown inhibits STING trafficking from the ER to ERGIC.

(a, b) HEK293T cells stably expressing non-targeting control (NT) shRNA or TMEM120A shRNAs were transiently transfected with plasmids expressing STING and GFP-ERGIC53 (green). 48h post transfection, cells were treated with 2 μg/ml cGAMP for 6h, then were fixed and permeabilized for immunostaining using STING (red) antibody followed by confocal analysis (a). STING-ERGIC53 co-localization was quantified using Pearson's correlation coefficient method. At least twenty cells expressing both STING and ERGIC53 were selected randomly for co-localization analysis by ImageJ (b). ERGIC53 is an ERGIC marker. Scale bar: 20 μm.

(c) HEK293T cells stably expressing non-targeting control (NT) shRNA or TMEM120A shRNAs were transiently transfected with plasmids expressing STING and GFP-ERGIC53. 48h post transfection, cells were treated with 2 μg/ml cGAMP for 6h, then collected for Flag-tag based immunoprecipitation and immunoblotting to detect ERGIC53 using GFP antibody. 10% of input was run and blotted. Asterisk: Flag-STING.

5. How was the infection dose (MOI = 5) of ZIKV – which seems very high – chosen for the screen?

Response: Since the screen is a survival screen aimed at discovering antiviral factors, the key is to minimize the background survival rate so that the surviving clones are indeed selected. Theoretically, MOI=5 would ensure over 99% cells are infected, leaving very few cells uninfected/survive by chance. A previous report showing ZIKV MR766 strain kills over 95% cells in 8 days at an MOI of 5 (Ref. S3). Based on these considerations, we infected Huh7 cells with ZIKV and conducted cell titer experiment. We also determined that approximately 95% of cells died from ZIKV-induced cytopathic

effects at day 10 post infection, consistent with the previous report. Therefore, we decided to use ZIKV at an MOI of 5 for our gain of function screening, which indeed enriched strong antiviral factors such as interferons and novel factors validated subsequently. We added a sentence on page 5, lines 5-6 to address this point.

References:

Ref. S3. Savidis, G. *et al.* Identification of Zika Virus and Dengue Virus Dependency Factors using Functional Genomics. *Cell Rep* **16**, 232-246, doi:10.1016/j.celrep.2016.06.028 (2016).

6. Fig 3d: luciferase is not a good control for the localization studies. It would be better to included proteins whose subcellular location has been well established as being ER-associate (+ ctrl) or restricted to some other organelle (e.g. MAVS with mitochondria/peroxisomes).

Response: Following the reviewer's suggestion, we have now performed immunofluorescence microscopy analysis to detect the co-localization of STING with TMEM120A with MAVS as a better control. We found that STING did not colocalize with MAVS in most cells, whereas TMEM120A co-localized with STING in most of co-transfected cells (Fig. S5 and also shown below). STING has also been described to be localized to the ER mostly and partially to the mitochondria in resting cells (**Ref. S4-6**). We have proved that TMEM120A is localized to the ER (Fig. 3b). Therefore, the new results further support our conclusion about the co-localization of STING and TMEM120A on the ER.

Figure S5. TMEM120A co-localizes with STING.

HEK293T cells were transiently transfected with plasmids expressing STING and HA-

Flag tagged MAVS, Calnexin or TMEM120A. 48h post transfection, cells were fixed and permeabilized for immunostaining using STING (red) antibody and Flag (green) antibody (a). STING-MAVS, Calnexin or TMEM120A co-localization was quantified using Pearson's correlation coefficient method. At least twenty cells expressing both STING and MAVS, Calnexin or TMEM120A were selected randomly for co-localization analysis by ImageJ software (b). Calnexin is an ER marker. MAVS is located on the mitochondria. Scale bar, 20 μ m.

References:

Ref. S4. Ishikawa, H. & Barber, G. N. STING is an endoplasmic reticulum adaptor that facilitates innate immune signalling. *Nature* **455**, 674-678, doi:10.1038/nature07317 (2008).

Ref. S5. Li, Z. *et al.* When STING Meets Viruses: Sensing, Trafficking and Response. *Front Immunol* **11**, 2064, doi:10.3389/fimmu.2020.02064 (2020).

Ref. S6. Zhong, B. *et al.* The adaptor protein MITA links virus-sensing receptors to IRF3 transcription factor activation. *Immunity* **29**, 538-550, doi:10.1016/j.immuni.2008.09.003 (2008).

7. Throughout the manuscript, the authors should make quantitative statements, e.g. "...significantly suppressed ZIKV infection [X fold]"(page 6, line 5)

Response: We thank the reviewer for the suggestion and have now we quantified significant changes and labeled them with [fold x] in the figures in revised manuscript.

8. Figures 3, 4, S5, S7: Please provide the Pearson's correlation coefficient for all the colocalization analysis included in this study.

Response: We thank the reviewer for the valuable suggestion. We have provided the Pearson's correlation coefficient for all the colocalization analysis in Figure 3, 4, S5, S8, S10 in revised manuscript.

9. Does TMEM120 expression/knock-down have any impact on other flaviviruses, in particular those that are sensitive to cGAS/STING signaling (e.g. DENV)

Response: As recommended by the reviewer, we determined the impact of TMEM120A expression on the infection of Dengue virus (DENV) and Yellow Fever Virus (YFV), two other flaviviruses. The results showed that TMEM120A expression indeed inhibited DENV and YFV infection in U87MG cells (Fig. S3). Furthermore, we also tested the effect of TMEM120A on HSV-1, a DNA virus, which is highly sensitive to cGAS/STING signaling. TMEM120A expression significantly inhibited HSV-1 infection in U87MG cells (Fig. S6a, b) and TMEM120A deletion remarkably increased HSV-1 infection in *Tmem120a* knockout MEFs (Fig. S6c, d).

Reviewer #2 (Remarks to the Author):

Li et al performed an overexpression screen in Huh7 cells to identify genes that inhibit Zika virus-induced cell death. Among the genes they identified was TMEM120A, a 6-transmembrane protein with poorly known function. RNAi of TMEM120A enhanced ZIKV replication. Knockdown of STING in U87MG cells abrogated the inhibitory effect of TMEM120A overexpression on ZIKV replication. TMEM120A was found to associate with STING on the ER and promote the translocation of STING from ER to ERGIC, a step previously shown to be important for STING signaling. Tmem120A-deficient mice had elevated ZIKV RNA and slightly weaker phosphorylation of TBK1 and IRF3 in response to cGAMP treatment.

Although the identification of TMEM120A as a potential regulator of STING trafficking may be interesting, there are several major concerns about the conclusions of the paper.

1) The entire paper relies on infection with ZIKV, which is an RNA virus, but the major function of STING is in immune defense against DNA viruses. If TMEM120A is important for STING signaling, a DNA virus such as HSV should be tested throughout.

Response: We thank the reviewer for raising this important point. To address the reviewer's concerns, we have performed the following experiments:

(1). We determined the effect of TMEM120A on HSV-1 infection and showed TMEM120A overexpression significantly inhibited HSV-1 infection in U87MG cells (Fig. S6a & S6b) and TMEM120A deletion remarkably promoted HSV-1 infection in *Tmem120a* knockout MEFs (Fig. S6c & S6d and also shown below).

Figure S6. TMEM120A inhibits HSV-1 infection.

(a, b) TMEM120A overexpression significantly inhibited HSV-1 infection in U87MG cells. U87MG cells stably expressing HA-Flag-tagged luciferase or TMEM120A were infected with HSV-1 at an MOI of 0.1 for 24h. Cells were then harvested and primers targeting gB and LAT were utilized to determine the RNA level (a) and DNA (b) of HSV-1 infection.

(c, d) *Tmem120a* deletion significantly promoted HSV-1 infection in MEFs. WT and *Tmem120a*^{-/-} MEFs were infected with HSV-1 at an MOI of 0.2 for 24h. Cells were then harvested and primers targeting gB and LAT were utilized to determine the RNA level (c) and DNA (d) of HSV-1 infection.

(2). We measured the phosphorylation of TBK1 after HSV-1 infection. We found that TMEM120A expression increased the phosphorylation of TBK1 after HSV-1 infection in U87MG cells (Fig. S13a) and TMEM120A deletion reduced the phosphorylated TBK1 in *Tmem120a* knockout MEFs (Fig. S13b and also shown below).

Figure S13. TMEM120A mediates HSV-1 induced phosphorylation of TBK1.

(a) TMEM120A overexpression increased the phosphorylation of TBK1 after HSV-1 infection in U87MG cells. U87MG cells stably expressing HA-Flag-tagged luciferase or TMEM120A were infected with HSV-1 at an MOI of 1 for 6h. Cells were collected for immunoblotting to detect the activation of TBK1 using phospho-TBK1 (p-TBK1), TBK1 antibodies. Ratio: p-TBK1/TBK1.

(b) *Tmem120a* deletion reduced the phosphorylated TBK1 induced by HSV-1 infection in MEFs. WT and *Tmem120a*^{-/-} MEFs were infected with HSV-1 at an MOI of 1 for 6h. Cells were collected for immunoblotting to detect the activation of TBK1 using p-TBK1, TBK1, antibodies. Ratio: p-TBK1/TBK1.

(3). Human TMEM120A overexpression also increased the transcription of cytokines and ISGs in HSV-1 infected RAW 264.7 cells, an immune cell line commonly used in HSV-1 study (Fig. 4k).

(4). As mentioned above in response to Reviewer 1's comment, we added DMXAA, another small molecule that stimulates STING-mediated DNA sensing signaling and found that TMEM120A potentiated the downstream signaling events (Fig. S15d).

2) The effect of TMEM120A deletion on the phosphorylation of TBK1 and IRF3 in response to cGAMP was quite modest (Fig 5e and 5f), especially considering that TMEM120A deletion also impaired cytokine induction by poly[I:C] (Fig 5g), which does

not activate STING. Thus, the effect of TMEM120A deletion might be non-specific (to the STING pathway).

Response: We agree with the reviewer's point. Actually, we repeated the results at least three times. The effect of TMEM120A deletion on the phosphorylation of TBK1 and IRF3 in response to cGAMP is highly reproducible. We looked for other reports about STING regulators such as STEEP (**Ref. S7**), SNX8 (**Ref. S8**), and found that the effect of TMEM120A deletion on the phosphorylation of TBK1 in response to cGAMP was similar with that of STEEP or SNX8 deletion as previously reported using the same quantification method through ImageJ. The quantification method is relative quantification, may not be very precise.

References:

- Ref. S7.** Zhang, B. C. *et al.* STEEP mediates STING ER exit and activation of signaling. *Nat Immunol* **21**, 868-879, doi:10.1038/s41590-020-0730-5 (2020).
- Ref. S8.** Wei, J. *et al.* SNX8 modulates innate immune response to DNA virus by mediating trafficking and activation of MITA. *PLoS Pathog* **14**, e1007336, doi:10.1371/journal.ppat.1007336 (2018).

Regarding poly (I:C), indeed, it has been reported that poly (I:C) does not induce Sting activation in MEFs (**Ref. S9-13**). But the authors transfected or stimulated cells with high concentration of poly (I:C) (>500ng/ml) in most of the reports. In our research, we transfected WT and *Tmem120a*^{-/-} MEFs with 300ng/ml poly (I:C) for 3h. To detect the specificity of TMEM120A to the STING pathway:

(1). We transfected plasmid expressing luciferase or TMEM120A into HEK293T cells overexpressing STING for 48h and then stimulated the cells with 300ng/ml poly (I:C) for 3h or 6h. Western blot result showed that TMEM120A overexpression enhanced poly (I:C)-induced phosphorylation of TBK1, which is dependent on STING overexpression (The result is shown below).

(2). We applied the same poly (I:C) stimulation to MEFs which were isolated from WT and *Sting*^{-/-} littermate (*Sting*^{fl}). The result showed that poly (I:C)-induced *Irf3* and *Isgs* had a lower level in *Sting*^{-/-} MEFs, which is consistent with the results we have got using *Tmem120a*^{-/-} MEFs (The result is shown below).

(3). We conducted RNA sequencing of WT and *Sting* KO MEFs which were transfected with 300ng/ml poly (I:C) for 6h. We found 149 genes significantly upregulated upon poly (I:C) treatment in WT MEFs and only 27 genes significantly upregulated in *Sting* KO MEFs, suggesting *Sting* may play an important role in *Isgs* expression induced by poly (I:C) (Data will be reported elsewhere).

(4). Based on the RNA sequencing data, we transfected WT and *Sting* KO MEFs with 300ng/ml or 600ng/ml poly (I:C) and detect the production of *Isgs* by RT-qPCR. We found that the transcription of *Isgs* were significantly impaired in *Sting* KO MEFs, especially in MEFs that were transfected with a lower poly (I:C) concentration (300ng/ml) (Data will be reported elsewhere).

These suggest that poly (I:C) treatment induces ISGs expression at least partially dependent on STING, and high poly (I:C) concentration would robustly induce RIG-I-like receptor (RLR) signaling which may overwrite the STING function. We will further investigate how poly (I:C) activates STING pathway based on our RNA sequencing data.

(a) TMEM120A promotes TBK1 phosphorylation in a STING-dependent manner. HEK293T cells expressing empty vector or STING were transiently transfected with luciferase or TMEM120A for 48h, then were treated with poly (I:C) for 3h or 6h. Cells were harvested for immunoblot. Ratio: p-TBK1/TBK1.

(b) *Sting* deletion weakens poly (I:C) induced type I IFN response in MEFs. WT and *Sting*^{-/-} MEFs were transfected with 300ng/ml poly (I:C) for 6h. Cells were harvested for RT-qPCR to detect the mRNA level of *Ifnb1* and other *Isgs*.

References:

- Ref. S9.** Abe, T. *et al.* Cytosolic-DNA-mediated, STING-dependent proinflammatory gene induction necessitates canonical NF- κ B activation through TBK1. *J Virol* **88**, 5328-5341, doi:10.1128/JVI.00037-14 (2014).
- Ref. S10.** Biolatti, M. *et al.* Human cytomegalovirus tegument protein pp65 (pUL83) dampens type I interferon production by inactivating the DNA sensor cGAS without affecting STING. *Journal of virology* **92** (2018).
- Ref. S11.** Franz, K. M. *et al.* STING-dependent translation inhibition restricts RNA virus replication. *Proc Natl Acad Sci U S A* **115**, E2058-E2067, doi:10.1073/pnas.1716937115 (2018).
- Ref. S12.** Ishikawa, H. *et al.* STING regulates intracellular DNA-mediated, type I interferon-dependent innate immunity. *Nature* **461**, 788-792,

doi:10.1038/nature08476 (2009).

Ref. S13. Konno, H. *et al.*. Cyclic dinucleotides trigger ULK1 (ATG1) phosphorylation of STING to prevent sustained innate immune signaling. *Cell* **155**, 688-698, doi:10.1016/j.cell.2013.09.049 (2013).

3) Many cancer cell lines do not express STING. Do Huh7 cells express STING? If not, the authors' model is called into question.

Response: We thank the reviewer for raising this important point. Although several reports have referred to the low level of endogenous level of STING in Huh7 cells (**Ref. S14, S15**), STING is indeed detectable as shown by several other research groups (**Refs. S16, S17**). We also detected STING expression in protein level in Huh7 cells which were utilized for our screening and virus infection, even though the expression is lower than two other cell lines.

STING is expressed in Huh7 cells. Huh7, HEK293T and HeLa cell lines were harvested for immunoblotting (STING antibody).

References:

Ref. S14. Thomsen, M. K. *et al.* Lack of immunological DNA sensing in hepatocytes facilitates hepatitis B virus infection. *Hepatology* **64**, 746-759, doi:10.1002/hep.28685 (2016).

Ref. S15. Liu, Y. *et al.* Hepatitis B virus polymerase disrupts K63-linked ubiquitination of STING to block innate cytosolic DNA-sensing pathways. *J Virol* **89**, 2287-2300, doi:10.1128/JVI.02760-14 (2015).

Ref. S16. Gan, E. S. *et al.* Dengue virus induces PCSK9 expression to alter antiviral responses and disease outcomes. *J Clin Invest* **130**, 5223-5234, doi:10.1172/JCI137536 (2020).

Ref. S17. Guo, F. *et al.* Activation of Stimulator of Interferon Genes in Hepatocytes Suppresses the Replication of Hepatitis B Virus. *Antimicrob Agents Chemother* **61**, doi:10.1128/AAC.00771-17 (2017).

4) Fig 3e: Is the association of STING and TMEM120A dependent on virus infection? A DNA virus should be tested in this case.

Response: We are grateful for the suggestion. To detect whether the association of STING and TMEM120A is dependent on virus infection, we infected U87MG cells overexpressing Flag-luciferase or Flag-STING with HSV-1 and performed

immunoprecipitation of Flag-STING using anti-Flag antibody. We confirmed that more TMEM120A was co-immunoprecipitated with STING after HSV-1 infection (Fig. S7 and also shown below). This indicates the binding of STING and TMEM120A in both infected and uninfected cells.

Figure S7. ZIKV and HSV-1 infection enhances the association of STING and TMEM120A.

U87MG cells expressing Flag tagged luciferase or STING and TMEM120A were infected with ZIKV or HSV-1 an MOI of 1 for 12h and then harvested for Flag-tag based immunoprecipitation and immunoblotting to detect the interaction between TMEM120A and STING or luciferase using TMEM120A antibody. 10% of input was run and blotted.

5) Fig 4h: the effect of TMEM120A overexpression on cGAMP-induced IRF3 phosphorylation was also very weak.

Response: We agree with the reviewer's criticism. The phosphor-IRF3 antibody is not as sensitive as phosphor-TBK1 antibody, which might contribute to the less significant changes detected. However, we have repeated the results for at least 3 times and found that the results are highly reproducible. Below we show the results from two other repeats. Importantly, the effect is consistent with increased TBK1 phosphorylation and transcriptional upregulation of cytokines and ISGs. Therefore, we conclude that this increase is meaningful and repeatable.

(a, b). TMEM120A promoted IRF3 phosphorylation in HEK293T cells. HEK293T cells stably expressing STING were transiently transfected with HA-Flag tagged luciferase or TMEM120A. 48h post transfection, these cells were stimulated with 2', 3'-cGAMP (2 μ g/mL) for the indicated time (0, 3h). Cells were collected for immunoblotting to detect the activation of IRF3 using p-IRF3, and IRF3 antibodies. Ratio: p-IRF3/IRF3.

REVIEWER COMMENTS

Reviewer #1 (Remarks to the Author):

The authors have adequately addressed the concerns that I had raised in the previous round of review.

Reviewer #2 (Remarks to the Author):

This revision has addressed most of my concerns. However, I was surprised to see that on page 13 of the rebuttal letter, there was a prominent expression of STING in HEK293T cells. Although there might be variations in different 293 cell clones, in most cases people do not see STING expression in HEK293T cells. This is consistent with the authors' having to use HEK293T-STING stable cells for their experiments. Since it's important to know whether Huh7 cells indeed express STING as indicated in my comments, the authors should repeat the experiment with knockout or knockdown controls to clarify their observation of STING expression in Huh7, HEK293T and HeLa cells.

I also remain concerned about Figure 5, where the authors showed that TMEM120A knockout only weakly inhibited p-TBK1 and p-IRF3 induced by cGAMP and also inhibited ISGs induced by poly[I:C]. This suggests a potential non-specific effect. If the authors claimed that poly[I:C] also activated STING, they should then pick another activator of the RIG-I pathway as a control.

Title: Gain-of-function genetic screening identifies antiviral function of TMEM120A by activating STING

Point by point responses to reviewers' comments:

We are delighted to know our revised manuscript has addressed the concerns of reviewer #1 and would like to thank reviewer #2 for the additional constructive comments on our manuscript. We have further revised the manuscript according to the comments and hope that the new revised manuscript could address all the concerns about our work.

Reviewer #2 (Remarks to the Author):

This revision has addressed most of my concerns. However, I was surprised to see that on page 13 of the rebuttal letter, there was a prominent expression of STING in HEK293T cells. Although there might be variations in different 293 cell clones, in most cases people do not see STING expression in HEK293T cells. This is consistent with the authors' having to use HEK293T-STING stable cells for their experiments. Since it's important to know whether Huh7 cells indeed express STING as indicated in my comments, the authors should repeat the experiment with knockout or knockdown controls to clarify their observation of STING expression in Huh7, HEK293T and HeLa cells.

Response: We appreciate the reviewer's comment. Following the comment, we have silenced STING using its specific siRNA in Huh7, HeLa and HEK293T cells and detected the protein levels of STING in these cells. We observed that STING knockdown significantly reduced the endogenous STING protein levels in Huh7, HeLa and HEK293T cells (Supporting Figure 1 below). This indicates that endogenous STING is indeed expressed in these cell lines which are used for our experiments.

Supporting Figure 1. STING is expressed in Huh7, HeLa and HEK293T cells which are used in our experiments. Huh7, HeLa and HEK293T cell lines were transfected with non-targeting (NT) or STING siRNA for 72h and then harvested for immunoblotting to measure protein level (STING antibody). LE, long exposure. SE, short exposure.

I also remain concerned about Figure 5, where the authors showed that TMEM120A knockout only weakly inhibited p-TBK1 and p-IRF3 induced by cGAMP and also inhibited ISGs induced by poly(I:C). This suggests a potential non-specific effect. If the authors claimed that poly(I:C) also activated STING, they should then pick another activator of the RIG-I pathway as a control.

Response: As recommended by the reviewer, we transfected WT and *Tmem120a*^{-/-} MEFs with 5' triphosphate hairpin RNA (3p-hpRNA, Invivogen), one of the well-characterized activator of the RIG-I pathway^{1,2}. We found that 3p-hpRNA treatment robustly induced *Ifnb* expression in MEFs (a shown in **Panel a** below) and *Tmem120a* deletion did not significantly affect the transcription of *Ifnb* and *Isgs* induced by 3p-hpRNA (**Panel b** shown below). These results indicate that 3p-hpRNA does not activate TMEM120a-STING signaling. Therefore, there might indeed be a non-specific effect of poly(I:C) as suggested by the reviewer. In light of the new data, we replaced the result of poly(I:C) treatment in Fig. 5g with the that of cGAMP treatment which showed more significant difference in ISG induction between WT and *Tmem120a*^{-/-} MEFs than that of poly(I:C). This removal does not affect the main conclusions of our study. We thank the reviewer for this very pertinent suggestion.

Supporting Figure 2. (a) 3p-hpRNA treatment robustly induced *Ifnb* expression in MEFs. WT and *Tmem120a*^{-/-} MEFs were transfected with 3p-hpRNA for 6h. Cells were harvested for RT-qPCR to detect the mRNA level of *Ifnb*.

(b) *Tmem120a* deletion did not affect 3p-hpRNA induced type I IFN response in MEFs. WT and *Tmem120a*^{-/-} MEFs were transfected with 3p-hpRNA for 6h. Cells were harvested for RT-qPCR to detect the mRNA level of *Ifnb* and other *Isgs*.

References:

- 1 Rehwinkel, J. *et al.* RIG-I detects viral genomic RNA during negative-strand RNA virus infection. *Cell* **140**, 397-408, doi:10.1016/j.cell.2010.01.020 (2010).
- 2 Hornung, V. *et al.* 5'-Triphosphate RNA is the ligand for RIG-I. *Science (New York, N.Y.)* **314**, 994-997, doi:10.1126/science.1132505 (2006).

REVIEWER COMMENTS

Reviewer #2 (Remarks to the Author):

I do not have additional concerns.

Title: Gain-of-function genetic screening identifies antiviral function of TMEM120A by activating STING

Point by point responses to reviewers' comments:

Reviewer #2 (Remarks to the Author):

I do not have additional concerns.

Response: We thank the reviewer for the acceptance of our manuscript.